

# Trends of inorganic and organic aerosols and precursor gases in Europe: insights from the EURODELTA multi-model experiment over the 1990–2010 period

Giancarlo Ciarelli[1,2,a,*], Mark R. Theobald[3], Marta G. Vivanco[3], Matthias Beekmann[1], Wenche Aas[4], Camilla Andersson[5], Robert Bergström[5,6], Astrid Manders-Groot[7], Florian Couvidat[2], Mihaela Mircea[8], Svetlana Tsyro[9], Hilde Fagerli[9], Kathleen Mar[10], Valentin Raffort[11], Yelva Roustan[11], Maria-Teresa Pay[12], Martijn Schaap[7], Richard Kranenburg[7], Mario Adani[8], Gino Briganti[8], Andrea Cappelletti[8], Massimo D'Isidoro[8], Cornelis Cuvelier[13], Arineh Cholakian[1,2], Bertrand Bessagnet[2,b], 10  Peter Wind[9,14] and Augustin Colette[2]

[1]Laboratoire Inter-Universitaire des Systèmes Atmosphériques (LISA), UMR CNRS 7583, Université Paris Est Créteil et Université Paris Diderot, Institut Pierre Simon Laplace, Créteil, France
[2]National Institute for Industrial Environment and Risks (INERIS), Parc Technologique ALATA, 60550 Verneuil-en-15  Halatte, France
[3]CIEMAT, Research Centre for Energy, Environment and Technology, Madrid, Spain
[4]Norwegian Institute for Air Research (NILU), Box 100, 2027 Kjeller, Norway
[5]Swedish Meteorological and Hydrological Institute, 60176 Norrköping, Sweden
[6]Department of Space, Earth and Environment, Chalmers University of Technology, 41296 Gothenburg, Sweden
[7]Netherlands Organisation for applied scientific research (TNO), P.O. Box 80015, 3508 TA Utrecht, the Netherlands
[8]Italian National Agency for New Technologies, Energy and Sustainable Economic Development (ENEA), Via Martiri di Monte Sole 4, 40129 Bologna, Italy
[9]Climate Modelling and Air Pollution Division, Research and Development Department, Norwegian Meteorological Institute (MET Norway), Blindern, 0313 Oslo, Norway
[10]Institute for Advanced Sustainability Studies (IASS), Postdam, Germany
[11]CEREA, Joint Laboratory Ecole des Ponts ParisTech – EDF R&D, Champs-Sur-Marne, France
[12]Barcelona Supercomputing Center, Centro Nacional de Supercomputación, Jordi Girona, 29, 08034 Barcelona, Spain
[13]ex European Commission - JRC, Ispra, Italy
[14]Faculty of Science and Technology, University of Tromsø, Tromsø, Norway
[a]Now at: Department of Chemical Engineering, Carnegie Mellon University, Pittsburgh, USA
[b]Now at: Hangzhou Futuris Environmental Technology Co. Ltd, Zhejiang Overseas High-Level Talent Innovation Park, No. 998 WenYi Road, 311121, Hangzhou, Zhejiang, China

[*]Corresponding author: Giancarlo Ciarelli (Giancarlo.Ciarelli@lisa.u-pec.fr)



**Abstract.** In the framework of the EURODELTA-Trends (EDT) modeling experiment, several chemical transport models (CTMs) were applied for the 1990–2010 period to investigate air quality changes in Europe as well as the capability of the models to reproduce observed long-term air quality trends. Five CTMs have provided modeled air quality data for twenty-one continuous years in Europe using emission scenarios prepared by IIASA/GAINS and corresponding year-by-year meteorology derived from ERA-interim global reanalysis. For this study, long-term observations of particle sulfate ($SO_4^{2-}$), total nitrate ($TNO_3$), total ammonium ($TNH_x$) as well as sulfur dioxide ($SO_2$) and nitrogen dioxide ($NO_2$) for multiple sites in Europe were used to validate the model results. The trends analysis was performed for the full twenty-one years (referred to as PT), but also for two 11-year sub-periods: 1990–2000 (referred to as P1) and 2000–2010 (referred to as P2).

The experiment revealed that the models were able to reproduce the faster decline in observed $SO_2$ concentrations during the first decade, i.e. 1990–2000, with a 64-76% mean relative reduction in $SO_2$ concentrations indicated by the EDT experiment (range of all the models) versus an 82% mean relative reduction in observed concentrations. During the second decade, P2, the models estimated a mean relative reduction in $SO_2$ concentrations of about 34-54%, which was also in line with that observed (47%). Comparisons of observed and modeled $NO_2$ trends revealed a mean relative decrease of 25% and between 19–23% (range of all the models) during the P1 period, and 12% and between 22-26% (range of all the models) during the P2 period, respectively.

Comparisons of observed and modeled trends in $SO_4^{2-}$ concentrations during the P1 period indicated that the models were able to reproduce the observed trends at most of the sites, with a 42-54% mean relative reduction indicated by the EDT experiment (range of all models) versus a 57% mean relative reduction in observed concentrations, and with good performances also during the P2 and PT periods. Moreover, especially during the P1 period, both modeled and observational data indicated smaller reductions in $SO_4^{2-}$ concentrations compared with its gas-phase precursor (i.e. $SO_2$), which could be mainly attributed to increased oxidant levels and pH-dependent cloud chemistry.

An analysis of the trends in $TNO_3$ concentrations indicated a 28-39% and 29% mean relative reduction in $TNO_3$ concentrations for the full period for model data (range of all the models) and observations, respectively. Further analysis of the trends in modeled $HNO_3$ and particle nitrate ($NO_3^-$) concentrations revealed that the relative reduction in $HNO_3$ was larger than that for $NO_3^-$ during the P1 period, which was mainly attributed to an increased availability of "free-ammonia". By contrast, trends in modeled $HNO_3$ and $NO_3^-$ concentrations were more comparable during the P2 period. Also, trends of $TNH_x$ concentrations were, in general, under-predicted by all models, with worst performance for the P1 period than for P2.

Trends in modeled anthropogenic and biogenic secondary organic aerosol (ASOA and BSOA) concentrations together with the trends in available emissions of biogenic volatile organic compounds (BVOCs) were also investigated. A strong decrease in ASOA was indicated by all the models, following the reduction in anthropogenic NMVOCs precursors. Biogenic emission data provided by the modeling teams indicated a few areas with statistically significant increase in isoprene emission and monoterpene emissions during the 1990–2010 period over Fennoscandia and Eastern European regions (i.e. around 14-27%), which was mainly attributed to the increase of surface temperature. However, the modeled BSOA concentrations did not linearly follow the increase in biogenic emissions. Finally, a comprehensive evaluation against positive matrix factorization (PMF) data, available during the second period (P2) at various European sites, revealed a systematic under-estimation of the modeled SOA fractions of between a factor of 3 to 11, on average, most likely because of missing SOA precursors and formation pathways, with reduced biases for the models that accounted for chemical aging of semi-volatile SOA components in the atmosphere.



## 1. Introduction

Particulate matter (PM) is one of the greatest environmental concerns, affecting climate and visibility, and having deleterious effects on human health (Cohen et al., 2017; Pope and Dockery, 2006; WHO, 2013). Although particulate matter can be directly emitted from different sources, e.g. power plants, industry and transport, PM with an aerodynamic diameter below 2.5 μm ($PM_{2.5}$) is mainly of secondary origin (Crippa et al., 2014), i.e. formed in the atmosphere after various reactions involving gas-phase precursors such as nitrogen oxide ($NO_2$), sulfur dioxide ($SO_2$), ammonia ($NH_3$), volatile organic compounds (VOCs) and several oxidants (e.g. OH, $O_3$ and $NO_3$). Particles in this size range can penetrate deeply into the respiratory system leading to respiratory and cardio-vascular problems. The formation mechanisms leading to the formation of secondary aerosols, especially the organic fraction, are complex, non-linear and still not fully understood (Bian et al., 2017; Lachatre et al., 2018; Tsigaridis et al., 2014).

Emissions of $SO_2$ and $NO_2$ have largely declined in Europe over the recent decades (Fagerli and Aas, 2008; Tørseth et al., 2012; UNECE LRTAP, 2016). For $SO_2$ and $NO_2$, emissions were reported to have declined by about 65% and 31%, respectively, between 1990 and 2009 whereas emissions of $NH_3$ were reported to have declined by about 29%, although the emission changes exhibit high spatial variability within the European domain (Tørseth et al., 2012). $NH_3$, which is emitted mainly from agricultural activities, is one of the key chemical species involved in the formation of secondary inorganic aerosol. It is the most important base in the atmosphere (Seinfeld and Pandis, 2012) and can react very rapidly with sulfuric acid ($H_2SO_4$), which is formed from the oxidation of $SO_2$ with OH (in the gas phase), $O_3$ and hydrogen peroxide ($H_2O_2$) in the aqueous phase, to form ammonium sulfate or ammonium bisulfate (Seinfeld and Pandis, 2012). If enough $NH_3$ is available after the neutralization of $H_2SO_4$, it can react with nitric acid ($HNO_3$), which is mainly formed form the oxidation of NO and $NO_2$, to produce the semi-volatile ammonium nitrate. Formation of ammonium nitrate usually occurs when the molar concentration of $NH_3 + NH_4^+$ is more than twice the sulfate concentration (i.e. "free ammonia regime") (Tsimpidi et al., 2007).

Past and future trends in the total PM concentration have recently obtained large attention thanks to the availability of long-term observational data-sets and increased computational power available for long-term CTM simulations. Tørseth et al. (2012) analyzed long-term air quality trends from the European Monitoring and Evaluation Programme (EMEP) during a period of 40 years. Their study showed a substantial reduction in ambient concentrations of sulfur species of about 70-90% since 1980, well in line with emission reductions, and a reduction of about 23% in $NO_2$ concentrations since the beginning of the 1990s. However, available observations of total nitrate ($TNO_3 = HNO_3(g) + NO_3^-(p)$) showed only a minor reduction (about 8%), compared to the larger reductions in $NO_2$. Aksoyoglu et al. (2014) performed an air quality modeling study with the Comprehensive Air Quality with extension (CAMx) model to evaluate air quality changes due to anthropogenic emission changes in the framework of the revised Gothenburg protocol. They performed air quality simulations for the emission years 1990, 2005 and 2020 with emission scenarios prepared from IIASA/GAINS. Their results indicated that the annual mean $PM_{2.5}$ concentration in Europe decreased by 20-50% between 1990 and 2005. Moreover, simulated annual mean $PM_{2.5}$ concentrations were 30% lower in 2020 compared with 2005, with larger decreases for eastern European countries (Aksoyoglu et al., 2014). Similarly, Colette et al. (2011) investigated the capability of six regional and global CTMs for simulating air quality changes between 1998 and 2007 with a focus on $NO_2$, $O_3$ and $PM_{10}$. Their results indicated that the models could reproduce the trends of primary pollutants, but they had difficulties in reproducing the small observed trends in $O_3$, and the year-to-year variability was under-estimated, in general. More recently, Banzhaf et al., 2015 applied the LOTO-EUROS model for the 1990–2009 period to investigate trends of air quality in Europe. They concluded that the model was able to well reproduce the observed trends in primary and secondary produced pollutants. In addition, they also performed a source apportionment study to evaluate the formation efficiency of secondary inorganic species during the 1990–2009 period. Their results





indicated an increase in $SO_4^{2-}$ formation efficiency (between 20–50%) as well as for $NO_3^-$ (up to 20%) compared with 1990.

Organic aerosol (OA) is often a major fraction of $PM_{2.5}$. OA is a complex mixture of thousands of organic compounds with different chemical and physical properties and volatilities (semi-volatile to low volatile) (Huang et al., 2014;

Jimenez et al., 2009). Numerous measurement campaigns performed in different parts of the world and periods of the year, have revealed that only a minor fraction of the observed total OA mass is directly emitted as primary organic aerosol (POA). A more abundant component, referred to as secondary organic aerosol (SOA), was found to often dominate the composition of OA especially in rural areas (Crippa et al., 2014).

The formation of SOA in the atmosphere is mainly initiated by the oxidation of gas-phase organic compounds in

different ranges of saturation concentrations, usually referred to as low-volatile, semi-volatile, intermediate-volatile and high-volatile ranges (Donahue et al., 2012, 2011). Some of the resulting gas-phase oxidation products will acquire lower saturation concentration due to the addition of oxygen-containing functional groups and will eventually condense on pre-existing organic particles leading to formation of SOA (depending on temperature and OA concentrations). On the other hand, other organic compounds will obtain lower molecular weight and will fall into higher saturation

concentration ranges through fragmentation, and they will likely reside in the gas-phase.

A recent model intercomparison exercise, AeroCom (Tsigaridis et al., 2014), investigated the performance of thirty-one global models with respect to OA revealing large differences between models in terms of SOA formation, mainly because of the assumptions made in the SOA scheme used (e.g. chemical aging, multiphase chemistry and semi-volatile SOA assumptions). In addition, comparison with several observational data-sets revealed that even though the models

were able to simulate the secondary nature of OA, they tended to largely underestimate the observed OA, especially in urban areas (Tsigaridis et al., 2014). In Europe, recent applications of CTMs have started to provide a comprehensive picture of the main sources of OAs as well as their temporal variation throughout the year. Bergström et al., 2012 applied the EMEP MSC-W model with the volatility basis set (VBS) model and tested different assumptions on the volatility distribution of POA as well as on the parameterizations of the aging processes. Their studies revealed an

underestimation of OA concentrations, especially during winter periods and in northern European countries, most likely as a result of uncertainties in the emissions from the residential sector (mainly wood burning emissions). Summer-time OA levels, on the other hand, were highly influenced by biogenic SOA precursors (isoprene and terpene), also confirmed by more recent studies (Cholakian et al., 2017; Chrit et al., 2017; Ciarelli et al., 2016).

In this study, we investigate the trends in SIA and SOA in Europe during the 1990–2010 period calculated by five

CTMs that participated in the EURODELTA-Trends exercise (Colette et al., 2017). The novel multi-model EURODELTA-Trends (EDT) exercise (launched within the Task Force on Measurement and Modelling of the EMEP Programme supporting the Convention on Long Range Transboundary Air pollution (CLRTAP)), provided 21-year of continuous $PM_{2.5}$ components and their gas-phase precursors concentrations over Europe from the year 1990, and with "real" year-to-year meteorological input data. It provides a base for validating the performance of multiple models over

an extended period (i.e. 1990–2010) and to assess the variation of various chemical species not routinely measured in Europe.

The paper is organized as follows: Section 2 provides a general overview of the EURODELTA-Trends experiment, with a description of the models participating in the exercise and the input data used to perform the experiment. The observational data is described in Section 2 along with information regarding the quality control criteria. Results and

discussions are presented in section 3. The trends in anthropogenic emissions and inorganic species are discussed in section 3.1 and section 3.2, respectively. An evaluation of the secondary organic aerosol fraction is presented in section 3.3 (for the 2000–2010 period) together with the trends in biogenic emission and anthropogenic and biogenic SOA



concentrations. Finally, conclusions are presented in Section 4.

## 2. Methods

### 2.1 Overview of the EURODELTA-Trends experiment

The EURODELTA-Trends experiment builds upon the expertise of the previous EURODELTA phases initiated in 2004 (van Loon et al., 2007). In the latest EURODELTA experiments, i.e. EURODELTA III, the performances of several CTMs were investigated for common air quality pollutants, i.e. $NO_2$, $O_3$, $SO_2$, and $PM_{10}$ and $PM_{2.5}$ at a European scale for specific periods of the EMEP and EUCAARI intensive measurements campaigns (Bessagnet et al., 2016).

The follow-up EURODELTA experiments, referred to as EURODELTA-Trends (EDT), aim at investigating the changes in air quality in Europe over the 1990–2010 period. In this framework, state-of-the-art CTMs, were applied over the European domain (Figure 1) with common input data (meteorological fields, anthropogenic emissions and boundary conditions). The participating models carried out extensive sensitivity tests that aimed at disentangling the role of different drivers (e.g. meteorology and emissions) on changes in air quality. The complete list of data available, chemical species and sensitivity tests are reported in detail in Colette et al., 2017.

In this study, one tier of simulations was used to investigate the models' capabilities to reproduce gas-phase PM precursors as well as SIA trends over the 1990–2010 period. This tier, referred to as tier 3A, provides twenty-one years of modeled air quality data in Europe driven with "real" meteorology, observation-based boundary conditions and anthropogenic emission scenarios based on the IIASA/GAINS model. Biogenic emissions were calculated separately by the different modeling teams using their own biogenic model driven by the meteorological data (e.g. temperature and radiation).

### 2.2 Description of the participating models

Eight state-of-the-art air quality CTMs delivered their results for the EDT experiments: CHIMERE (Mailler et al., 2017; Menut et al., 2013), CMAQ (Byun and Schere, 2006), EMEP MSC-W (Simpson et al., 2012), LOTOS-EUROS (Manders et al., 2017; Schaap et al., 2008), MATCH (Andersson et al., 2015, 2007; Robertson et al., 1999), MINNI (Mircea, 2016), Polyphemus (Mallet et al., 2007; Sartelet et al., 2012) and WRF-Chem (Grell et al., 2005; Mar et al., 2016). Given the large computational demand of the simulations, only five modeling teams were able to deliver 21 years of continuous air modeled data: CHIMERE, EMEP MSC-W, LOTOS-EUROS, MATCH and MINNI, the results of which are used in this study. Most of the other models provided air quality data for 3 intermediate years: 1990, 2000 and 2010.

The set-up for each participating model is reported in Table 1. The complete list of physical and chemical schemes (including dry and wet deposition parameterizations) can be found in (Colette et al., 2017). The models differ in terms of the adopted gas-phase chemistry mechanisms as well as SIA and SOA formation modules. Here, we briefly describe the main characteristics of the various schemes used by the models.

Various gas-phase schemes were used to perform the gas-phase chemistry (Table 1): The Carbon Bond mechanism version 4 (referred to as TNO-CBM-IV), EmChem09, MELCHIOR2 and SAPRC99.

- The TNO-CBM-IV gas-phase scheme (Schaap et al., 2009), used by the LOTOS-EUROS model, includes 33 gas-phase species and 9 organic species emitted directly into the atmosphere. Most of the included organic species are lumped according to the carbon-carbon bond type and only a minority of them are explicitly represented (e.g. isoprene and formaldehyde). 104 chemical reactions and 14 photolytic reactions are mapped



in the TNO-CBM-IV mechanism for gas-phase chemistry.

- The EmChem09 gas-phase scheme (Simpson et al., 2012), used by EMEP MSC-W and MATCH models, include 72 species, 137 chemical reactions and 26 photochemical reactions. The rates and products were designed to be as close as possible to the IUPAC recommendations (http://www.iupac-kinetic.ch.cam.ac.uk/) and most of the reaction coefficients were taken from Atkinson et al., 2006, 2004. The MATCH model used a modified version of isoprene chemistry based on the work of Carter, 1996 and Langner et al., 1998.

- The MELCHIOR2 gas-phase scheme (Derognat et al., 2003), used by the CHIMERE model, is a reduced version of the MELCHIOR1 mechanism and it includes 120 chemical reactions and hydrocarbon degradation as in the EMEP gas-phase mechanism, with a few adaptations included for low-NOx conditions and NOx-nitrate chemistry. All rate constants are taken from Atkinson et al., 1997 and De Moore et al., 1994.

- The SAPRC99 gas-phase scheme (Carter, 2000), used by the MINNI model, includes a detailed speciation of about 400 types of VOCs and with detailed reaction schemes for most of the non-aromatic hydrocarbons and oxygenates in the presence of NOx. The isoprene photooxidation is explicitly included, the "four-product" condensed isoprene mechanism considers methacrolein, methyl vinyl ketone, lumped C5 unsaturated aldehyde products (ISOPROD), and the methacrolein PAN analogue (MPAN).

To resolve the composition and phase state of inorganic aerosol, most of the models used the ISORROPIAv2.1 scheme (version 1.7 for the MINNI model and version II for LOTOS-EUROS) which assumes thermodynamic equilibrium with its gas phase precursors (Nenes et al., 1999, 1998). The EMEP MSC-W model adopted the approach proposed by Binkowski and Shankar, 1995, i.e. the MARS equilibrium module, and does not include sodium chloride and dust components, whereas the MATCH model is based on the work of Mozurkewich, 1993. Transformation of $HNO_3$ to coarse nitrate is included by all the models except MINNI.

As already mentioned, $NH_3$ is a key ingredient for the formation of secondary inorganic aerosols. $NH_3$ compensation points are included in LOTOS-EUROS to account for the presence of $NH_3$ in the stomata, external leaf surfaces or at the soil surface and partially included in the EMEP MSC-W model by assuming zero $NH_3$ dry deposition to growing crops.

Different gas-phase and thermodynamic organic aerosol schemes with various levels of complexity were used by the modeling teams (Table 1): The volatility basis set with and without aging of SOA (Bergström et al., 2012a; Simpson et al., 2012), referred to as VBS-NPAS and VBS-NPNA, respectively, the $H^2O$ mechanism (Couvidat et al., 2012) coupled with the SOAP module (Couvidat and Sartelet, 2015), and the SORGAM mechanism (Schell et al., 2001). None of the models included emission of semi-volatile organic compounds (SVOCs) and/or of intermediate-volatile organic compounds (IVOCs). LOTOS-EUROS did not enable any SOA scheme and therefore the organic model description is not included here.

- The VBS-NPAS and VBS-NPNA organic aerosol modules, used by the EMEP MSC-W and MATCH models respectively, assumes primary organic aerosol (POA) emission to be non-volatile, assuming European emission inventories to consist of inert PM compounds. Semi-volatile SOA is formed from oxidation of anthropogenic and biogenic volatile organic compounds (VOC) (for details regarding the volatility basis set and SOA-yields see Bergström et al., 2012). In the EMEP (VBS-NPAS) model the OH-reaction rate for SOA



aging is set to $4.0\times10^{-12}$ cm$^3$ molecule$^{-1}$s$^{-1}$; each reaction of the organic compounds in the gas-phase decreases the volatility by one order of magnitude and increases the mass by +7.5 % to account for oxygen-addition (fragmentation processes are not included). SOA aging is not included in the VBS-NPNA scheme.

-      The H$^2$O organic aerosol module (Couvidat et al., 2012), used by the CHIMERE model, uses different types of surrogate organic species: hydrophilic species (which condense preferentially into an aqueous phase) and hydrophobic species (which condense only into an organic phase). These surrogate species are produced from the oxidation of volatile organic compounds. In H$^2$O, SOA are formed from 4 classes of precursors: aromatic compounds, isoprene, monoterpenes and sesquiterpenes. For aromatic compounds, toluene and xylene are used
as SOA precursors when reacting with the OH radical and without accounting for SOA aging. The H$^2$O mechanism accounts for the effect of nitrogen oxides on SOA formation as well as the dissociation of organic acids in an aqueous phase, the oligomerization of aldehydes. More details of the scheme can be found in Couvidat et al., 2018, 2012.

-      The SORGAM mechanism (Schell et al., 2001), used by the MINNI modelling system, includes 4 SOA precursors classes (alkanes, alkenes, aromatics and monoterpenes) to represent the contributions of anthropogenic precursors and biogenic precursors to SOA formation. The Volatile Organic Compounds (VOCs), are oxidized by reactions with hydroxyl radical (OH), ozone (O$_3$), and nitrate radical (NO$_3$). The anthropogenic SOA are formed from aromatics like toluene, xylene, cresol, from internal alkenes and long
"alkanes" as those grouped together in the ALK5 and OLE2 classes, respectively, in SAPRC99 gas-phase mechanism.  Biogenic SOA is produced only by monoterpenes which partitioning parameters are obtained from a weighted average of smog chamber experiments for α-pinene, β-pinene, d3-carene, sabinene, limonene.

### 2.3 Emissions

#### 2.3.1    Biogenic and natural emissions

Emissions of biogenic volatile organic compounds (BVOCs) were not prescribed by the EDT experiments. Each participating team used their own emission model to calculate biogenic emissions.
One group of models used the MEGANv2.04 (Guenther et al., 2006) and MEGANv2.1 (Guenther et al., 2012) emission models: CHIMERE and MINNI, respectively. CHIMERE uses highly resolved spatiotemporal data (30 arcsec every 8 days) generated from MODIS for Leaf area index (LAI) inputs. The 30 arcsec USGS (US Geophysical Survey) land-use database is used to provide information on the plant functional type (PFT). The PFT is then combined with the emission factors for each functional type of Guenther et al. (2012) to compute the landscape average emission factors. MINNI
derived them from the CORINE Land Cover (CLC2006) inventory. The MEGAN model is driven with meteorological variables, such as temperature, wind speed, humidity, solar radiation and soil moisture. The leaf area index retrieved from the TERRA/MODIS satellite is used to simulate the vegetation growth (eight-day and one-month average LAI data at 0.25° × 0.25° degrees resolution for CHIMERE and MINNI, respectively). Common BVOCs species such as isoprene, α-pinene as well as other classes of monoterpenes, are generated for each hour and grid cell of the domain. In
the CHIMERE model, emissions of sesquiterpenes are also included and used as an input for SOA chemistry. More information on the MEGAN emission algorithms can be found in Guenther et al., 2006, 2012.
The second group of models: LOTOS-EUROS, MATCH, EMEP MSC-W used a detailed tree inventory of 115 species for 30 European countries based on the work of Koeble and Seufert, 2001 and aggregated tree species based on land-



cover types. For this group of models, the environmental factors to derive biogenic emissions include the light correction factor ($\gamma_L$) and the temperature correction function ($\gamma_T$), which are applied to three types of emission categories: isoprene, pool-dependent monoterpenes and light-dependent monoterpenes based on (Guenther et al., 1993). More information on the EMEP MSC-W BVOC emission algorithm can be found in (Simpson et al., 2012).

Finally, sea-salt, emitted in water droplets from the sea during high wind speed conditions and as results of breaking of waves and/or bursting of air bubbles, are included in all the models, based on different schemes, as described in Colette et al., 2017. Windblown dust emission were taken into account by all the models except MATCH, while road traffic dust resuspension was only included in the EMEP MSC-W model (Colette et al., 2017).

### 2.3.2    Anthropogenic emissions

Anthropogenic gridded emissions by country and activity, i.e. SNAP (Selected Nomenclature for reporting of Air Pollutants) codes, were estimated using the Greenhouse gas - Air Pollution Interactions and Synergies (GAINS) model (Amann et al., 2011). Emission of $SO_x$, $NO_x$, $NH_3$, non-methane volatile organic compounds (NMVOCs) as well as primary $PM_{2.5}$, $PM_{10}$, black carbon and primary organic aerosol were prepared at a 0.25° × 0.40° resolution (latitude ×

longitude). Anthropogenic emissions were calculated for the years 1990, 1995, 2000, 2005 and 2010, and linearly interpolated by country and activity sector for the 5-years periods to obtain the continuous 21-year emission dataset. Data for the different emission sectors were obtained from Eurostat (http://ec.europa.eu/eurostat), the International Energy Agency (IEA, 2012) and the UN food and Agriculture Organization (FAO) (http://www.fao.org/statistics/en/). Additionally, data from the International Fertilizer Association (IFA) and the COPERT model (Athanasiadis et al.,

2009) were used for the agriculture and transportation sectors, respectively. An error in primary particulate emission matter for Russia, North Africa and maritime areas for the period 1991-1999 was identified at the end of the exercise. However, the effect of the error was estimated to be very limited (Theobald et al., 2019).

The complete anthropogenic emission dataset accounts for source-specific emission limits as well as for various European air quality directives (e.g. the UNECE Gothenburg Protocol, UNECE, 1999). This emission dataset, referred

to as ECLIPSE_V5, was delivered by IIASA as country national totals by activity sector. It was subsequently spatialized by INERIS on the EURODELTA-Trends grid for use in the CTMs using the gridding process described in Terrenoire et al., 2015 and Bessagnet et al., 2016. For the residential heating sector (SNAP2) a proxy based on population density was applied using a bottom-up inventory available for France. More information about the re-gridding can be found in (Colette et al., 2017).

### 2.3.3    Meteorological data

To provide meteorological inputs to the modeling teams, dynamically downscaled regional climate model simulations was used in combination with ERA-interim global reanalysis data (Dee et al., 2011). The Weather Research and

Forecast Model (WRF version 3.3.1; Skamarock et al., 2008) was used at a resolution of 0.44°to generate the meteorological parameters. To reduce the uncertainty of the meteorological data, WRF was re-run with ERA-interim reanalysis data in grid-nudging mode as described in Stegehuis et al., 2015 and subsequently interpolated at a 25 km resolution to match the EDT grid, although there were a few differences between the procedures of the modeling team. LOTOS-EUROS used RACMO2-downscaled data and MATCH used HIRLAM-downscaled data. More information on

the meteorological inputs can be found in (Colette et al., 2017).





### 2.3.4 Observational data and trend assessment

The observations are reported to EMEP, and the original time series are available in EBAS (http://ebas.nilu.no). The datasets chosen for the trend assessment have passed the completeness criteria of 75% of data available over the full 1990–2010 period and had undergone visual screening tests. The secondary dataset with annual and seasonal average concentrations is available from the webpage set up by Task Force on Measurements and Modeling (TFMM) for this study (https://wiki.met.no/emep/emep-experts/tfmmtrendstations). The datasets include yearly measurements of long-term air concentrations of sulfur dioxide ($SO_2$), particle sulfate ($SO_4^{2-}$), nitrogen dioxide ($NO_2$), total nitrate ($TNO_3 = HNO_3(g) + NO_3^-(p)$) and total ammonium ($TNH_x = NH_3(g) + NH_4^+(p)$) performed in Europe between 1990 and 2010. Overall, the numbers of observational sites available for each of the species are 30, 20, 25, 13 and 16 for $SO_2$, $SO_4^{2-}$, $NO_2$, $TNO_3$ and $TNH_x$, respectively. Figure 2 illustrate the geographical distribution of the observational sites for each of the species, all classified as rural background stations. It can be noted that most of the stations are located over the Northern and Central part of the domain. The complete list of the observational sites is reported in Table S1.

$NO_2$ is mainly sampled with the manual method where $NO_2$ is selectively absorbed on impregnated glass sinters. Some sites do however use chemiluminescence monitor with molybdenum converter, which is not selective for $NO_2$, thus these measurements might be biased, and this is especially important in areas with low concentrations (Reed et al., 2016), but it is not assumed that the trends will be largely affected when same method is used during the whole period. The other components are mostly measured using a filterpack sampler with no size cut off in the inlet. The 3-stage filterpack separates gas and aerosol species, but for nitrogen compounds this separation might be biased due to the volatile nature of $NH_4NO_3$. Therefore $TNO_3$ and $TNH_x$ are usually used for robust estimate of the atmospheric nitrogen loading (Tørseth et al., 2012). However, it is recommended to report the measurements of all the species since it may give valuable insight into the gas/particle ratio despite possible biases. Details of the method used are found in the annual data report (i.e. EMEP, 2012 for the 2010 data).

The linear trends for each species and observational site were calculated with the Theil-Sen method (Sen, 1968) and their significances were evaluated at 95% confidence level ($p < 0.05$) using the non-parametric Mann-Kendall test (Kendall, 1948; Mann, 1945). Trends in observational data were compared with trends in modeled data calculated with the same methodology. Since anthropogenic emissions did not decline linearly during the full period covered by the experiment (1990–2010, referred to as PT), and larger emissions reduction are expected during the earlier 90s, the trend analysis was performed for two sub-periods: a first period between 1990 and 2000, referred to as P1, and a second period between 2000 and 2010, referred as to P2. In addition, to provide a more comprehensive picture of the trends in the air pollutant concentrations, the trends analysis was also performed for several sub-regions adapted from the commonly used PRUDENCE climatic zones classification (http://ensemblesrt3.dmi.dk/quicklook/regions.html). The extension of the sub-regions used in the study is reported in Figure 1.

The evaluation of modeled SOA was performed using an extensive data-set of secondary organic aerosol concentrations retrieved with Positive Matrix Factorization (PMF) analysis (Paatero, 1999) and recently compiled by Tsimpidi et al., 2016. This data-set includes SOA average concentrations at 51 stations in Europe during the P2 period. In order to remove local pollution events, likely not included in emission inventories, stations with average SOA concentrations higher than 7 µg m$^{-3}$ during the measurement period were excluded from the analysis (i.e. 3 stations). Most of the measurements were performed during short campaigns using aerosol mass spectrometers (AMS) in different periods of the years, lasting from about two weeks to one month. The spatial distribution of the stations is presented in Figure S1. The complete list of stations used is reported in Table S2 along with information regarding the year and the seasons during which measurements were made.



### 3. Results and Discussion

#### 3.1 Trends in anthropogenic emissions

Table 2 reports the absolute and relative trends in $SO_x$, $NO_x$, $NH_3$ and NMVOCs emissions for the full 1990–2010 period as well as for the P1 and P2 periods over the entire domain. For the full period, $SO_x$ emissions show a decline of about 69%. $SO_x$ emissions declined faster during the P1 period compared with the P2 with decreases of 54 and 37%, respectively (Table 2). The large reduction in $SO_x$ emissions was largely attributed to emission reductions in the "combustion in energy and transformation industries" sector, largely achieved by the switch to low-sulfur containing

fuels (e.g. natural gas) and the adoption of desulphurization technologies in large industries.

$NO_x$ emissions were reduced by 25% during the P1 period and by 17% during the P2 period. These reductions were mainly achieved through emission reductions in the road transport sector following the introduction of the new EURO standards for passenger cars. However, in 2010 this sector still represented the most important source of anthropogenic $NO_x$ emissions in Europe (EEA, 2012). Important $NO_x$ emission reductions were also achieved thanks to the adoption

of low-$NO_x$ burners and selective and non-selective catalytic reduction measures for the "combustion in energy and transformation industries" sector.

$NH_3$ emissions declined only a little compared to $SO_x$ and $NO_x$ emissions. $NH_3$ emission mainly arises from agricultural activities, which had less stringent controls compared to $SO_x$ and $NO_x$ emission ceilings. $NH_3$ emissions declined by 19% over the P1 period but only by 6% over the P2 period.

Emissions of NMVOCs showed a decline of 59% over the full 1990–2010 period with similar relative reductions achieved during the P1 and P2 periods: 33% per period. NMVOC emission reductions were mainly driven by the road transport sector, and by the year 2010, most of the NMVOC emissions arise from the use of solvents (EEA, 2012). Huang et al., 2017 compiled a global gridded data set of speciated NMVOC emissions for the 1970–2010 period and analyzed the trends. Among the different world regions, North America and Europe were reported to have reduced their

NMVOC emissions since 1970 due to the introduction of EURO emission standards for vehicles. A significant reduction of formaldehyde emissions was reported in 2010 compared with 2000, mainly because of the increasing adoption of EURO standards and the transition from coal to cleaner fuels (e.g. natural gas). The latter resulted in a substantial decrease in the aromatic species and in an increase in the contribution of alkanes and alkanals to the emissions of NMVOCs (Huang et al., 2017).





### 3.2 Trends in inorganic species

#### 3.2.1 Comparison of modeled and observed $SO_2$ and $NO_2$ concentrations trends

5    Figure 3 and Table 3 report the mean relative trends of all the sites included in the analysis (Table S1).

Overall, the observations indicate relative reductions of 25, 12 and 36% in $NO_2$ concentrations for the P1, P2 and PT, respectively, with the models estimating similar ranges of relative reductions, i.e. 19–23% for the P1 periods, 22–26% for the P2 period and 44–47% for the full period, depending on the model (Table 3). Only about half of the individual observed trends were reproduced within a factor of two by individual models and all models performed worse in the second period (P2), overestimating the observed trends (Figure 4 and Table 3). Such behavior could indicate possible difficulties for CTMs in capturing long-term trends at relatively low concentrations with small annual changes, typical of the P2 period, it also could point to overestimated negative trends in national emission data bases.

For $SO_2$, the observed relative reductions were 82, 47 and 97% for the P1, P2 and full periods, respectively (Figure 3 and Table 3). The models indicate very similar ranges of $SO_2$ reductions, i.e. 64-76% for the P1 periods, 34-54% for the P2 period and between 84-97% for the full period, depending on the model (Table 3). This is in line with the emission reduction trends presented in Section 3.1 and with previous trend studies for Europe (Tørseth et al., 2012). Table 3 also reports the fraction of model estimates within a factor of 2 of the observed trends. Most of the models were able to reproduce the observed $SO_2$ trends within a factor of two at most of the individual sites (Figure 3 and Table 3). Overall, model performance was better during the P1 period compared with P2.

20    Figure 5 shows the percentage of statistically significant/non-significant increasing/decreasing observed and modelled $SO_2$ and $NO_2$ trends:

For $SO_2$, most of the stations had significant decreasing trends in concentrations during the P1 period, with only a small fraction of the stations with non-significant decreasing trends. All the models were able to reproduce this pattern, albeit with a slight over-estimation of the significant-decreasing fraction. During the P2 periods, there is a tendency of most of the models to over-estimate the number of significant trends, with CHIMERE and LOTOS-EUROS being closer to the fraction of significant/non-significant decreasing trends indicated by the observations. As expected, the agreement between the modeled and observed fractions of significant/non-significant increasing/decreasing trends was improved for the full period (PT), mainly because of the larger number of data-points in the time series, with all the sites indicating significant observed and modeled decreasing trends.

30    For $NO_2$, the models were able to reproduce the observed fraction of significant/non-significant increasing/decreasing trends in the P1 period, with most of the models indicating a significant-decrease in $NO_2$ concentrations at most of the stations (slightly lower for EMEP MSC-W). The analysis for the P2 period shows that the reduced fraction of observed significant-decreasing trends compared with the P1 period, was not well reproduced by the models, all of them tending to over-estimate the fraction of significantly decreasing trends (Figure 5). Again, a possible explanation for the degraded model performance during the P2 period could be related to the relatively low pollutant concentrations, which might be challenging to model at such coarse resolution, as well as to uncertainties in the measurements data (see Section 2.3.4). One site in Ireland (IE0001R) was the only site with significant increasing observed trends during the P1 and PT periods, a result which was not reproduced by any of the models.

#### 3.2.2 Comparisons of modeled and observed $SO_4^{2-}$, $TNO_3$ and $TNH_x$ concentration trends

Figure 6 shows the mean modeled and observed relative trends in $SO_4^{2-}$, $TNH_x$ and $TNO_3$ for all the sites included in the analysis (Table S1). In consistency with the gas-phase analysis, the trends are reported for the two sub-periods, i.e.



1990–2000 (P1) and 2000–2010 (P2), as well as for the full period (PT).

Overall, the observations indicated that concentrations of $SO_4^{2-}$ declined by 57, 14 and 66% for the P1, P2 and PT periods (mean of all the stations, Figure 6 and Table 4), with the models indicating relative reductions of 42−54% for the P1 periods, 23−35% for the P2 period and 61−78% for the full period, depending on the model. The reductions in
$SO_4^{2-}$ concentrations were larger during P1 than during the P2 period and most of the model estimates were within a factor of two of the observed values for all the periods (Figure 7 and Table 4). Two sites, one in Ireland (IE0001R) and one in Poland (PL0003R), showed an increase in $SO_4^{2-}$ concentrations (Figure 7), which none of the models were able to reproduce.

The percentage of statistically significant/non-significant increasing/decreasing trends in the observed and
modeled $SO_4^{2-}$ trends is reported in Figure 8, showing a good agreement between the observed and modeled significances (and their direction) for the three periods. Statistically significant increasing trends in $SO_4^{2-}$ concentrations were only observed at the PL0003R site in Poland during the P2 period (Figure 8), a result which none of the models were able to reproduce. Interestingly, observed $SO_4^{2-}$ concentrations declined less than those of $SO_2$ (Table 3), a behavior also reproduced by all the models. The non-linear dependencies between the reduction in $SO_2$ and $SO_4^{2-}$
concentrations are influenced by different factors First, the strong reduction in $SO_x$ emissions will increase the availability of OH radicals, which will directly enhance the homogeneous reaction rate of $SO_2$. Second, all the models account for the dependence of the aqueous chemistry of $SO_2$ on pH levels. Thus, heterogeneous reactions of $SO_2$ are also expected to proceed more efficiently due to the increase of pH levels over time.

Observations of $TNH_x$ reveal that concentrations declined by 28, 22 and 46% for P1, P2 and PT, respectively (Table 4).
In general, most of the models under-predict the relative changes; the modeled relative reductions for the P1, P2 and PT periods were 15–26%, 14–21% and 27–38%, respectively, with the P1 period showing only a minor fraction of the data-points within a factor of two (Figure 7 and Table 4). Indeed, large uncertainties remain in terms of ammonia emissions, which might affect model performance for $TNH_x$. Moreover, we would like to underline that none of the participating model accounted for the influence of meteorology (e.g. temperature) on ammonia emissions and relied on static
emission profiles provided by the EURODELTA exercise. Recent studies, however, have shown that better agreement in terms of the modeled ammonia concentrations can be achieved when ammonia emissions are modulated with local meteorological conditions (Backes et al., 2016; Hendriks et al., 2016). Compared with the other investigated species, a larger variation in terms of the significance of the trends can be seen in Figure 8, with most of the models tending to over-estimate the fraction of significant decreasing trends. Statistically significant increasing trends in observed $TNH_x$
concentrations were found at one station in Norway (NO0039R) for the full period (PT), with none of the models being able to reproduce this feature.

$TNO_3$ concentrations, on the other hand, declined to a lesser extent than those of $SO_4^{2-}$ and $TNH_x$. For all periods, the observed relative changes in $TNO_3$ concentration were 16, 19 and 29%, for the P1, P2 and PT periods, respectively, with the models estimating similar ranges for the P1 and P2 periods, i.e. 16-19%, 8-27% and 28-39%, for the P1, P2 and
PT periods, respectively (Figure 6 and Table 4). Most of the model estimates were more than a factor of two larger than the observed values for the P1 and PT periods (Figure 7 and Table 4). The percentage of statistically significant/non-significant increasing/decreasing observed and modeled $TNO_3$ trends revealed that most of the models were able to reproduce the large fraction of non-significant decreasing observed trends. The EMEP and MATCH models estimate a larger fraction of significant decreasing trends than the other models in both the P1 and P2 periods, where CHIMERE
and MINNI show the largest fraction of non-significant decreases. CHIMERE also shows the largest fraction of non-significant increasing trends during the P2 period (Figure 8).



### 3.2.3    Trends in modeled HNO₃ and NO₃⁻ concentrations for different sub-regions

In order to further investigate the trends in $TNO_3$ concentrations described in the previous paragraph, we also

investigated the modeled trends in $HNO_3$ and $NO_3^-$ concentrations (for the different sub-regions in Figure 1).

Figure 9 illustrates the relative trends in $HNO_3$ and $NO_3^-$ (sum of the coarse and fine particle fractions) for the P1 (first two columns in Figure 9) and P2 (last two columns in Figure 9) periods, for all the models that participated in the experiment. In general, during the P1 period, the models indicate larger significant decreases in $HNO_3$ compared with $NO_3^-$, especially over the Fennoscandia and central European regions. A few differences in the spatial distribution of the

modeled $HNO_3$ and $NO_3^-$ trends can be seen in Figure 9. For instance, for the LOTOS-EUROS model, the significant relative trends in both $HNO_3$ and $NO_3^-$ concentrations are more comparable during the P1 period whereas MINNI and CHIMERE estimates larger areas of non-significant trends in $NO_3^-$ concentrations, with some significant-increases in few parts of the domain. Figure 10 and Table 5 show the models' average relative changes and standard deviation (over land) of $HNO_3$ and $NO_3^-$ for all the PRUDENCE zones for the P1 and P2 periods. During the P1 period, the regions

classified as Alps (AL), British Island (BI), Benelux area (BX), France (FR) and Mid-Europe (ME), had the largest decrease in $HNO_3$ concentrations, with average decreases between 36 and 44% (Table 5). A comparison of $HNO_3$ and $NO_3^-$ relative trends for the same regions shows that $NO_3^-$ concentrations declined to a lesser extent, i.e. around 20%, which is roughly half of the modeled relative reduction in $HNO_3$ concentrations. The largest difference between the relative reduction in $HNO_3$ and $NO_3^-$ concentrations occurred over the Scandinavian regions for the P1 period, of 24 and

5%, respectively. On the other hand, the reduction of $HNO_3$ and $NO_3^-$ concentrations were comparable in the Eastern-European region and over the Iberian Peninsula (Table 5), as well as during the P2 periods.

The non-linear response of $HNO_3$ concentrations and $NO_3^-$ concentrations, i.e. larger relative reduction in $HNO_3$ compared to $NO_3^-$, could be attributed to the shift in the thermodynamic equilibrium of $HNO_3$ versus particle nitrate $NO_3^-$. In fact, for specific regions and especially during the P1 period, the large reduction in $SO_2$ emissions increased the

availability of "free-ammonia" and thus the transfer of more $HNO_3$ into the particle phase, favoring the formation of ammonium nitrate. This also increases the $TNO_3$ lifetime as dry deposition is much more rapid for $HNO_3$ than for $NO_3^-$. This effect could contribute to the reduced $TNO_3$ decreases with respect to the $NO_2$ decrease. Figure 11 shows the modeled $HNO_3/NO_3^-$ and $NH_3/NH_4^+$ molar ratio for the P1 and P2 periods, as predicted by all the models. In general, the models indicate significant decreasing trends in the $HNO_3/NO_3^-$ ratio especially over Scandinavian regions (to a

lesser extent in the LOTOS-EUROS model) and a strong increase in the $NH_3/NH_4^+$ ratio over the whole domain except for some Eastern European areas.

### 3.2.4    Trends in modeled SO₄²⁻ and SO₂ concentrations for different sub-regions

In this section the trends in modeled $SO_4^{2-}$ and $SO_2$ for the sub-regions reported in Figure 1, are discussed. Figure 12 illustrates the relative trends in $SO_2$ and $SO_4^{2-}$ concentrations for the P1 (first two column) and P2 (last two columns) periods for all the models that participated in the experiment. Trends were predicted to be statistically significant over the whole domain during the P1 period, and to a lesser extent over Eastern and Norther European regions during the P2 period. A larger decline in $SO_2$ concentrations during the P1 period was predicted in the Mid-Europe areas, around 85%

relative reductions, compared to the Iberian Peninsula, Mediterranean and Fennoscandia areas (average reduction 24, 48 and 33%, respectively) (Figure 12 and Table 6). During the P2 period, the modeled relative reductions of $SO_2$ were in the range 33-65%, with the Iberian Peninsula showing a larger reduction compared to the P1 period. As already discussed in the evaluation section, $SO_4^{2-}$ declined to a lesser degree than $SO_2$, by 34 and 59% in the P1 period and 30-





49% in the P2 period, likely because of the increased availability of oxidant species and pH-dependent cloud chemistry.



### 3.3 Trends in organic species

#### 3.3.1 SOA evaluation

During the P2 period, various field campaigns were performed in Europe using aerosol mass spectrometer instruments (AMS) to measure ambient OA concentrations at various sites. Using factor-analysis techniques, i.e. positive-matrix factorization analysis (PMF), it was possible to apportion the measured OA to a direct emitted organic factor, i.e. POA, and a more oxidized secondary factor, referred to as SOA. Even though this methodology is affected by various sources of uncertainties, we used this dataset to provide a general benchmark for modeled SOA performance during the P2

period.

   Figure 14 shows the average modeled and observed SOA concentrations (retrieved from PMF analysis) for all the sites included in the analysis (Table S2 and Figure S1), as well as for campaigns carried out during winter and summer periods (astronomical seasons). All the models underestimate observed SOA concentrations, in general, with larger variabilities between models than for the secondary inorganic species (section 3.2.2). On average, the models

underestimated the SOA concentrations by about a factor of 3 to 11 (Table 6, all periods), depending on the specific model (MATCH underestimating the most and EMEP MSC-W the least) and with a larger underestimation during winter periods. The EMEP MSC-W model which accounts for aging of SOA, was closer to the observations, with average SOA concentrations of about 0.7 µg m$^{-3}$. The higher SOA mass modeled by the EMEP MSC-W model could be explained by the shift of relatively high-volatile organic compounds toward lower volatility ranges when aging

processes are accounted for (with a reaction rate toward OH of $4.0\times10^{-12}$ cm$^3$ molecule$^{-1}$s$^{-1}$ in the case of the EMEP MSC-W model for both ASOA and BSOA). Such processes will increase the SOA mass since low-volatile oxidation products will rapidly condense into the particle-phase. Interestingly, the MATCH model which used the same VBS scheme as the EMEP MSC-W model, but without considering SOA aging processes, tends to under-estimate SOA concentrations substantially (Figure 14). This indicates the importance of these chemical mechanisms in CTMs and

their impact on SOA formation. On the other hand, the models based on the two-product scheme and molecular surrogate approach scheme, i.e. MINNI and CHIMERE, respectively, yielded very similar results for the total SOA mass, with SOA concentrations ranging in between the two VBS models (i.e. around 0.4 and 0.6 µg m$^{-3}$, averaged over all sites).

   Figure S2 and Figure S3 illustrate the modeled and observed (retrieved with PMF analysis) SOA concentrations at the

individual sites for winter and summer periods. In general, the models had difficulties in reproducing the SOA concentrations at specific urban sites, such as Paris and Manchester (in both summer and winter periods), and in reproducing high levels of SOA concentrations at few specific sites, e.g. Harkingen and Payerne, where emissions from biomass burning are high. This could be due to missing aerosol precursors (SVOCs emissions) in the resident-heating sectors, which have been shown to have high uncertainties (Denier van der Gon et al., 2015).

Figure 15 shows the modeled relative and absolute contributions of anthropogenic and biogenic secondary organic aerosols to SOA concentrations. For most of the models, larger contributions of ASOA to SOA were estimated during winter periods and/or in urban areas (e.g. in Paris, Manchester and Payerne), whereas the BSOA contribution to SOA was largest during warmer periods. Especially in summer, large emissions of biogenic volatile organic compounds can act as an important source of SOA. The CHIMERE model simulated the largest contribution of BSOA, with only minor

variations between the stations and periods.



### 3.3.2    Trends in BVOCs emissions and SOA concentrations

In this section, the trends in BVOCs emissions, i.e. isoprene and terpene, are presented together with the trends in
BSOA and ASOA concentrations. The trends analysis for BSOA and ASOA is reported for the full 1990–2010 period,
for the different PRUDENCE zones and with the methodology as defined in Section 2.3.4. Note that not all the
participants provided biogenic emissions for the full 21 years period, and only the EMEP MSC-W, CHIMERE and
MATCH models provided year-by-year emissions of isoprene and monoterpenes for the EDT experiments. Moreover,
for the EDT set-up, CHIMERE does not include biogenic emissions for Latitudes north of 65°N.

Figure 16 and Figure 17 illustrate the trends in isoprene and monoterpenes (first two columns) for the full 1990–2010
period. An increase in both isoprene and monoterpene species was found especially over Eastern Europe (EE), with
relative increases of a 15–27% for isoprene and 14–18% for monoterpenes, and over the Fennoscandia regions (SC),
with relative increases of 12–24% in isoprene emissions and 7–17% in monoterpene emissions, depending on the model
(Figure 16 and Figure 17). Interestingly, the  increase in biogenic emissions was predicted by all biogenic models (i.e.,
MEGANv2.1 and the one using vegetation data from Koeble and Seufert, 2001). These increases were mainly attributed
to the increase in surface temperature used to drive the different biogenic models (Figure S5), which however were
found to be not significant. The increase in surface temperature was found to be larger over Fennoscandia,
Mediterranean and Easter European areas compared to the remaining zones, (i.e. increases around 0.02, 0.02 and 0.03 K
$y^{-1}$, respectively). We want to underline that the surface solar radiation (SSR) could also plays an important role for the
emission of biogenic precursors (especially for isoprene). The strong reduction in $SO_4^{2-}$ concentrations described in
section 3.2.2, especially in the eastern regions of the domain, might have induced an increase in the incoming solar
radiation, referred to as brightening-periods (Wild 2009, 2011), which could affect the emission of biogenic species.
However, none of the models that participated in the exercise have explicitly accounted for such an interaction and the
ERA-interim forcing data rely mainly on climatological aerosol profiles. On the other hand, recent sensitivity studies
performed in Europe showed that such effects might be relatively small (Oikonomakis et al., 2018). Figure 16 and
Figure 17 also show the relative trends in BSOA and ASOA concentrations for all the models that were able to provide
twenty-one years of data. Even though some models indicated few increases in the biogenic SOA concentrations over
the Fennoscandia regions, these increases were found to be smaller than the increase in biogenic emissions and in some
cases, biogenic SOA concentrations were also estimated to have declined (Figure 16 and Figure 17). This might sound
counterintuitive, since one would expect more biogenic SOA to be produced as more biogenic precursors are available
and, in general, the increased availability of OH radicals due to the reduction in $NO_x$ and $SO_x$ emissions, thus increasing
the oxidation efficiency of biogenic SOA precursors, especially for isoprene. A possible explanation for this non-linear
relation between the trends in biogenic emissions and the trends in BSOA concentrations could be due to the trends in
the anthropogenic OA concentrations. As shown in Figure 16 and Figure 17 a strong decrease in the ASOA
concentrations was found for the entire 1990–2010 period, in line with the reduction in the NMVOC precursors
described in Section 3.1 (mainly from transportation sectors). Modeled ASOA concentrations indicate a decline of
about 60–70% over the whole domain (considering only land areas). This might have had on effect on the formation of
the BSOA fraction: in fact, the strong reduction in ASOA concentrations, and other aerosol organic and inorganic
components, will reduce the availability of organic and inorganic material onto which the low-volatile oxidized
compounds can condense, directly affecting the formation of the BSOA fraction. Additionally, oxidant levels and thus
oxidation pathways could have changed over time, affecting as said before OH, but also $NO_3$ concentrations.



## 4. Conclusions

A modelling experiment to evaluate the capability of several chemical transport models (CTMs) to reproduce long-term air quality trends in Europe was initiated within the EURODELTA-Trends (EDT) exercise.

Common anthropogenic emissions, meteorological input data and boundary conditions were used by the participants, whereas the chemical and physical parameters varied between the models. Modeled air quality data for the 1990–2010 period was validated against quality-controlled long-term measurements with a focus on several primary and secondary inorganic and organic pollutants.

In general, the experiment revealed that the models were able to reproduce relatively well the observed trends in gas-

phase precursors (i.e. $SO_2$ and $NO_2$), as well as secondary inorganic species, i.e. sulfate ($SO_4^{2-}$), total nitrate ($TNO_3$) and total ammonium ($TNH_x$), with a few exceptions at some specific sites. The range of modeled trends over 1990–2010 encompass the observed ones for $SO_2$, $SO_4^{2-}$ and $TNO_3$, but not for $TNHx$ and $NO_2$. The modeled relative declines of $NO_2$ concentrations were found to be 19–23% during the 1990–2000 period (P1) and 22–26% during the 2000–2010 period (P2), depending on the model. These values were in-line with the relative trends calculated from the

observations, around 25 and 12%, for the P1 and P2 periods, respectively (mean values of all sites) even if models did not catch the observed stronger decrease in P1 and weaker decrease in P2. Their difficulty in reproducing the weaker decline over the second period is attributed to the challenge in modelling low $NO_2$ levels at EMEP background sites during that period.

The large decline in $SO_2$ and $SO_4^{2-}$ concentrations during the early 90s, due to the switch to low-sulfur fuels (e.g.

natural gas) and the adoption of desulfurization technologies, was well reproduced by the models, with most of the absolute trends in observations being reproduced within a factor of 2. As expected, $SO_2$ decreases faster than $SOx$ emissions, and $SO_4^{2-}$ decreases less. This is due to the increase in cloud pH that accelerates in-cloud sulfur chemistry that consumes $SO_2$ to form $SO_4^{2-}$, constituting a positive retroaction for $SO_2$ decrease and a negative retroaction for $SO_4^{2-}$. This effect was well reproduced by the models.

$TNHx$ decreases much faster than $NH_3$ emissions (respectively 46% and 15% over 1990–2010). This is due to a change of $TNHx$ partitioning, which shifts towards gas when the atmospheric load of acids ($H_2SO_4$ and $HNO_3$) decrease. Consequently, a larger fraction of $TNHx$ is in $NH_3$ form, which deposits faster than $NH_4^+$, leading to a positive retroaction in enhancing the downward trend of $TNHx$. Deposition plays a critical role in $TNHx$ trends, and it has been noted that the observed decrease in wet deposition in the 1990s was largely driven by a couple of monitoring stations

that experienced a sharp drop between 1995 and 1996 (Theobald et al., 2019) which the models fail to capture.

The trends in $TNH_x$ concentrations were thus under-estimated, especially during the first decade (1990–2000), with the models exhibiting larger discrepancies compared to the other investigated species. Ammonia emissions certainly play an important role in the model performance for $TNH_x$. In fact, large uncertainties remain regarding present-day ammonia emissions and higher uncertainties are probably to be expected during the early 1990s.

The models estimated relatively lower trends in $TNO_3$ concentrations compared to other inorganic species (during the P1 periods), which was also indicated by the observations. A further analysis of the modeled $HNO_3$ and $NO_3^-$ components revealed that $HNO_3^-$ declined more than $NO_3^-$ during the 1990–2000 period. We attributed the later to a possible shift in the thermodynamic equilibrium of $HNO_3$ following the strong reduction in $SO_2$ concentrations, resulting in more "free-ammonia" available to drive the $HNO_3$ into the particle phase. Such an effect was particularly

enhanced over Fennoscandia regions where differences up to a factor of 5 in the modeled relative reductions of $HNO_3$ and $NO_3$ concentrations were found, 24% relative reduction in $HNO_3$ and 5% for $NO_3^-$, respectively (average values of all the models for the P1 period). Because $HNO_3$ deposits faster than $NO_3^-$, the shift of $TNO_3$ partitioning towards particles increases the lifetime of atmospheric nitrogen (as reported by Simpson et al., 2014), which contributes to



explain that TNO$_3$ decreases less than NOx emission.

A comprehensive data-set of SOA concentrations retrieved from positive matrix factorization analyses (PMF) was used to investigate the models' capabilities of reproducing the SOA concentrations during the 2000–2010 period. The analysis of modeled SOA concentrations indicated that the models under-estimated the SOA fraction by varying extents, by a factor of 3 to 11, depending on the model, suggesting that large uncertainties in the SOA formation mechanisms as well as in the emissions of SOA precursors remains, and more studies are needed to better elucidate the evolution of the SOA fraction, especially with a long-term prospective. The under-estimation of the SOA fraction seemed to be more pronounced during winter periods, in line with previous studies indicating missing SOA precursors in the residential sector, one of the major contributors to SOA concentrations in Europe during winter periods. Therefore, this experiment confirmed once more the need to improve emission inventories of primary organic aerosol for the residential sector, especially regarding wood-burning emissions.

The analysis of the modeled trends in emissions of biogenic volatile organic compounds (BVOCs), isoprene and monoterpene, revealed an increase in these precursor emissions during the 1990–2010 period, especially in Eastern European regions and in Fennoscandia regions, by about 20%. The increase was independent of the land-use and biogenic model used and was mainly attributed to the increase of the surface temperature during the 1990–2010 period. Modeled trends in ASOA concentrations indicated a strong reduction following emission reductions of non-methane volatile organic precursors, by around 60%, because of the implementation of new EURO standards for passenger cars, among other. However, modeled trends in BSOA concentrations remain less clear. Despite the modeled increase in biogenic emissions, modeled BSOA concentrations showed relatively small increasing trends, or even decreasing trends. A possible explanation was mainly attributed to the reduction in the aerosol mass indicated by all the models. The latter could eventually reduce the condensation sink of low-volatility organic compounds and reduce the capability to form additional organic material from biogenic precursors, despite the increase in BVOCs emissions. Thus, more work is still needed to better characterize the trends in organic aerosol, and especially of the BSOA fraction.





**Data and code availability.** Technical details of the EURODELTA project simulations that permit the replication of the experiment are available on the wiki of the EMEP Task Force on Measurement and Modelling (https://wiki.met.no/emep/emep-experts/tfmmtrendeurodelta, last access: 21 December 2018), which also includes ESGF links to corresponding input forcing data. The EURODELTA-Trends model results are made available for public

use on the AeroCom server (https://wiki.met.no/aerocom/ user-server, last access: 21 December 2018). See Colette et al. (2017) for full terms and conditions for the use of these data. The R code and procedures used for the model evaluation are available from the corresponding author upon request.

**Author contributions.** ACo coordinated the EURODELTA-Trends (EDT) exercise and WA was responsible for the

compilation and quality control of the observations. The following modelling teams set-up, pre-processed, ran and post-processed the simulations for each model: FC, BB, MGV and ACo for CHIMERE; ST, HF and PW for EMEP; AM, MS and RK for LOTO-EUROS; CA and RB for MATCH; MM, MA, GB, ACa and MD for MINNI. Additional post-processing of model output and uploading to the AeroCom server was done by KC. All of the analyses presented in this paper were carried out by GC with assistance from MT, KM, VR, YR, MTP, ACo, ACh and MB.

**Competing interests.** The authors declare that they have no conflict of interest.

**Acknowledgments.** G. Ciarelli was supported by ADEME in the frame of the MISTRALS/ChArMEx project and the Swiss National Science Foundation (grant no. P2EZP2_175166). The Ineris coordination of the Eurodelta-Trend

exercise was supported by the French Ministry in charge of Ecology in the context of the Task Force on Measurement and Modelling of the EMEP Programme of the LRTAP Convention. Meteorological forcing with the WRF model were provided by R. Vautard and A. Stegehuis from LSCE/IPSL. The CHIMERE simulations where performed using the TGCC super computers under GENCI computing allocation. The participation of CIEMAT was financed by the Spanish Ministry of Agriculture and Fishing, Food and Environment. The MATCH participation was partly funded by

the Swedish Environmental Protection Agency through the research program Swedish Clean Air and Climate (SCAC) and partly by NordForsk through the research programme Nordic WelfAir (grant no. 75007). The computing resources and the related technical support used for MINNI simulations have been provided by CRESCO/ENEAGRID High Performance Computing infrastructure and its staff. The infrastructure is funded by ENEA, the Italian National Agency for New Technologies, Energy and Sustainable Economic Development and by Italian and European research

programmes (http://www.cresco.enea.it/english). MINNI participation to this project was supported by the "Cooperation Agreement for support to international Conventions, Protocols and related negotiations on air pollution issues", funded by the Italian Ministry for Environment and Territory and Sea. The simulations with the EMEP MSC-W model were supported by the Research Council of Norway in the framework of the Programme for Supercomputing: through the EMEP project (grant NN2890K) for CPU, and the Norstore project "European Monitoring and Evaluation

Programme" (grant NS9005K) for data storage. The GAINS emission trends were produced as part of the FP7 European Research Project ECLIPSE (Evaluating the Climate and Air Quality Impacts of Short-Lived Pollutants); grant no. 282688.




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

WHO, 2013. Health risks of air pollution in Europe – HRAPIE – Summary of recommendations for question D5 on "Identification of concentration-response functions" for cost-effectiveness analysis.




**Figures and Tables.**

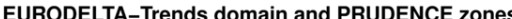

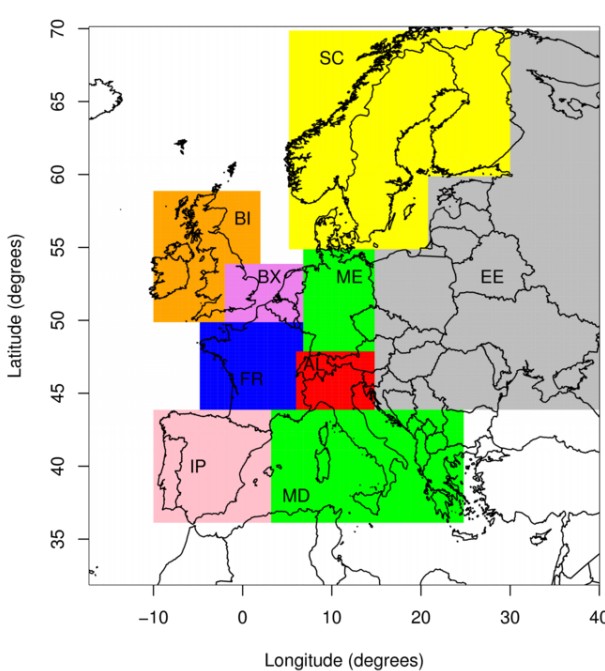

**Figure 1:** The extension of the EURODELTA-Trends domain as well as of the sub-regions adapted from the PRUDENCE zones. From South to North: Mediterranean regions (MD), Iberian Peninsula (IP), France (FR), Alps (AL), Mid-Europe (ME), Eastern Europe (EE), Benelux regions (BX), British Isles (BI) and Fennoscandia (SC).





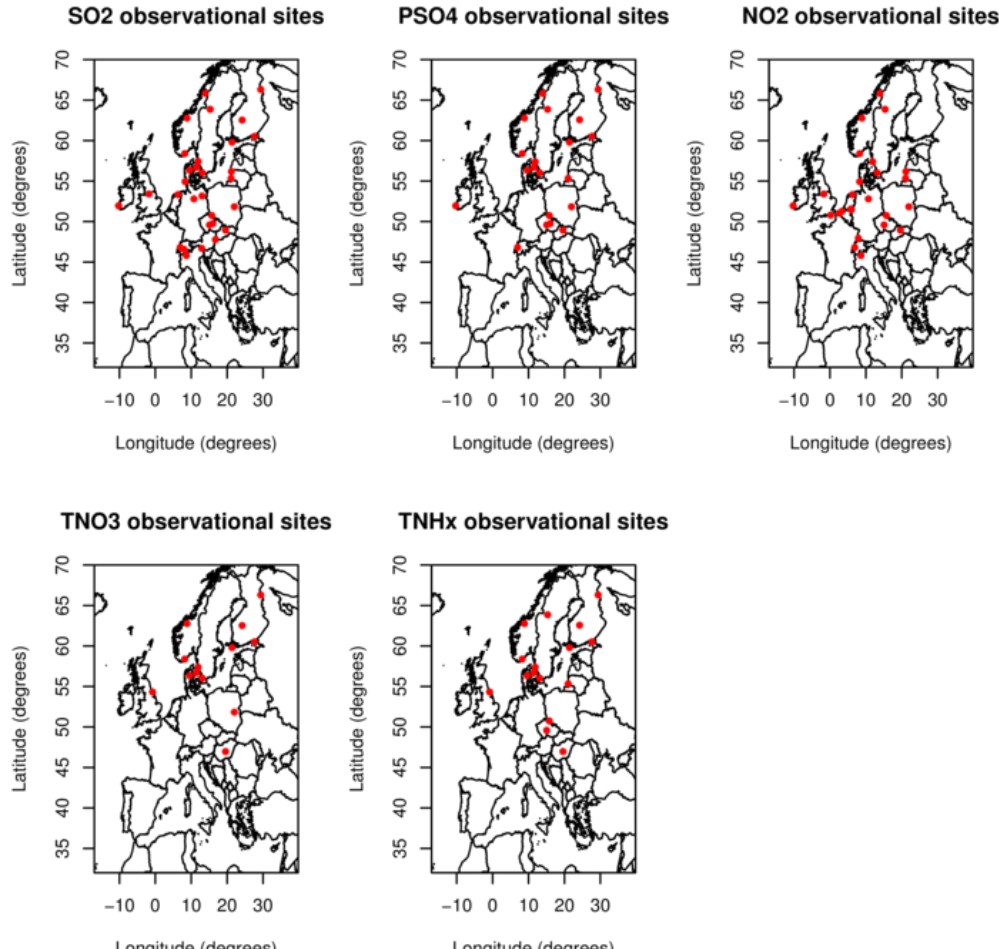

**Figure 2:** Locations of the observational sites (red dots). The numbers of observational sites available for each species are 30, 20, 25, 13 and 16 for $SO_2$, $SO_4^{2-}$ (PSO4), $NO_2$, $TNO_3$ and $TNH_x$, respectively.




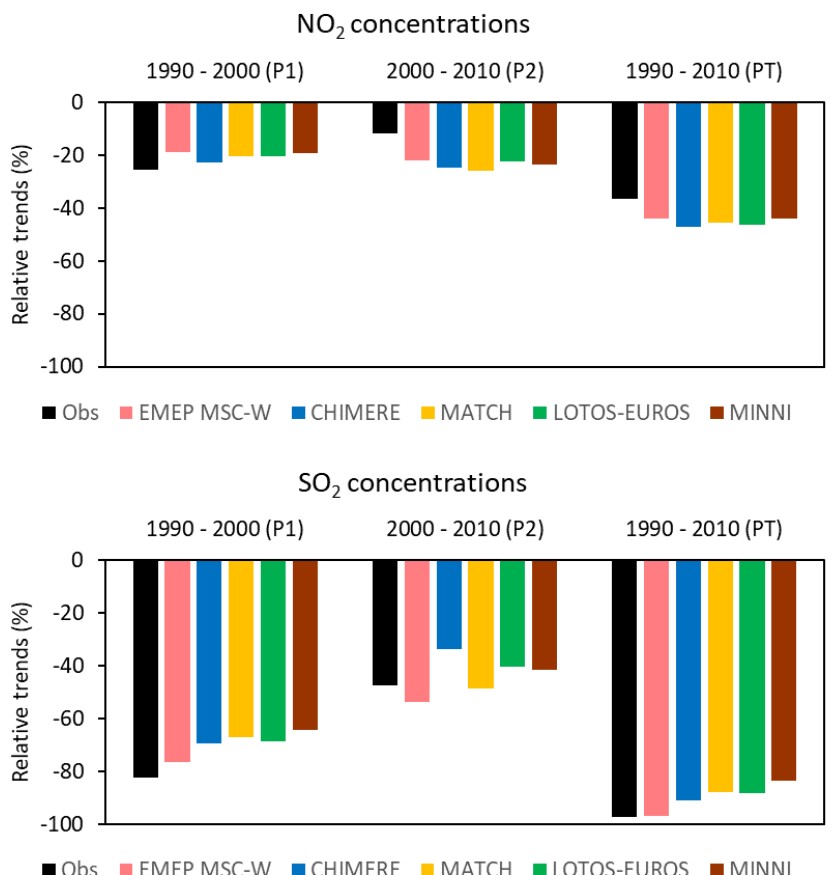

**Figure 3:** Modeled and observed (Obs) mean relative trends of $NO_2$ (upper-panels) and $SO_2$ concentrations (lower-panels) for the P1 (1990–2000), P2 (2000–2010) and PT (1990–2010) periods.





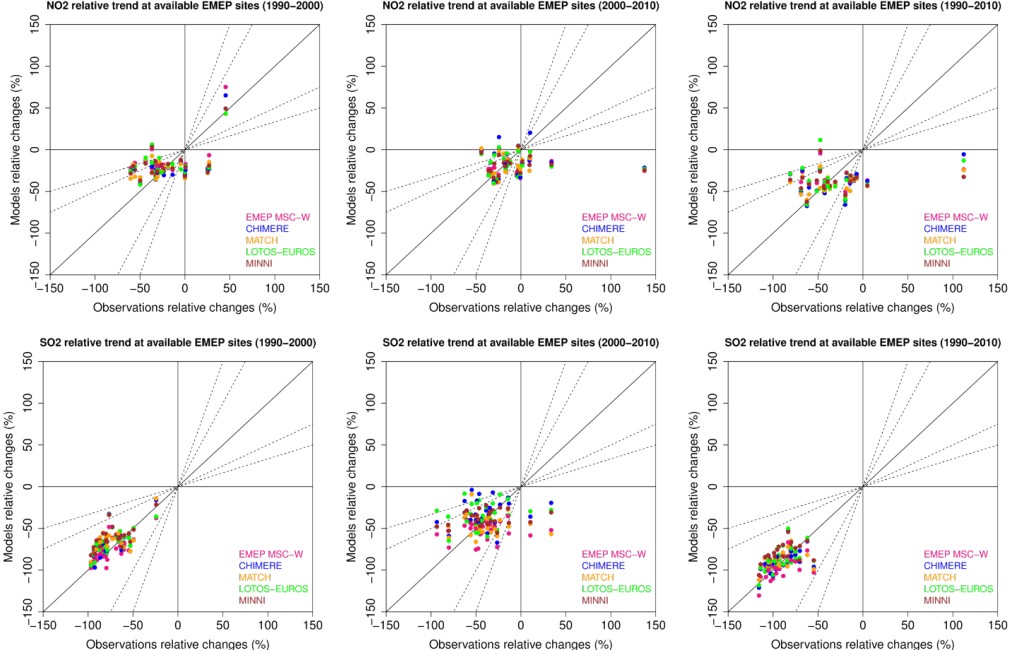

**Figure 4:** Modeled and observed NO$_2$ (upper-panel) and SO$_2$ (lower-panel) relative trends for the P1 (1990–2000), P2 (2000–2010) and PT (1990–2010) periods (left to right). The continuous line indicates the 1:1 line, and the dotted lines indicate the 1:2 and 1:3 lines (and their reciprocals).





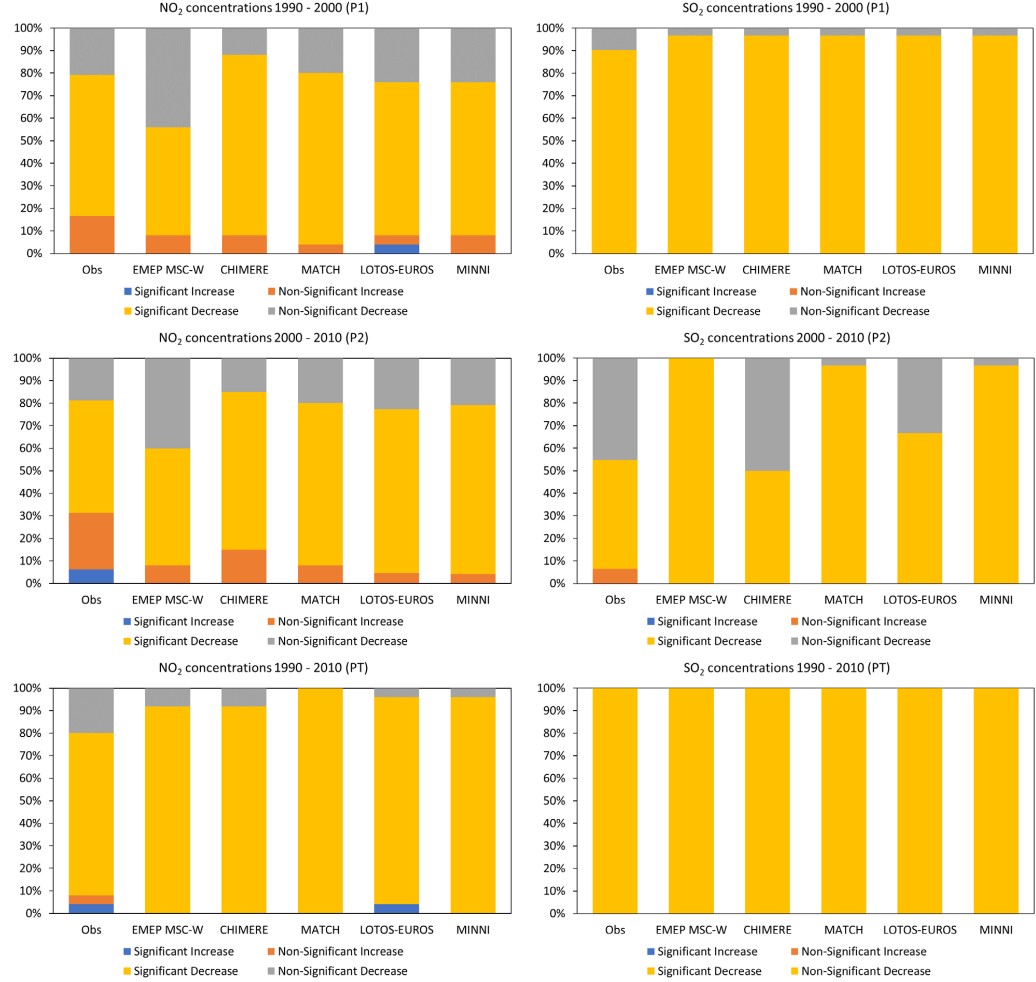

**Figure 5:** Percentage of statistically significant/non-significant increasing/decreasing trends in the observations and modeled data for NO₂ (left) and SO₂ (right) for the P1 (1990–2000), P2 (2000–2010) and PT (1990–2010) periods (from top to bottom).




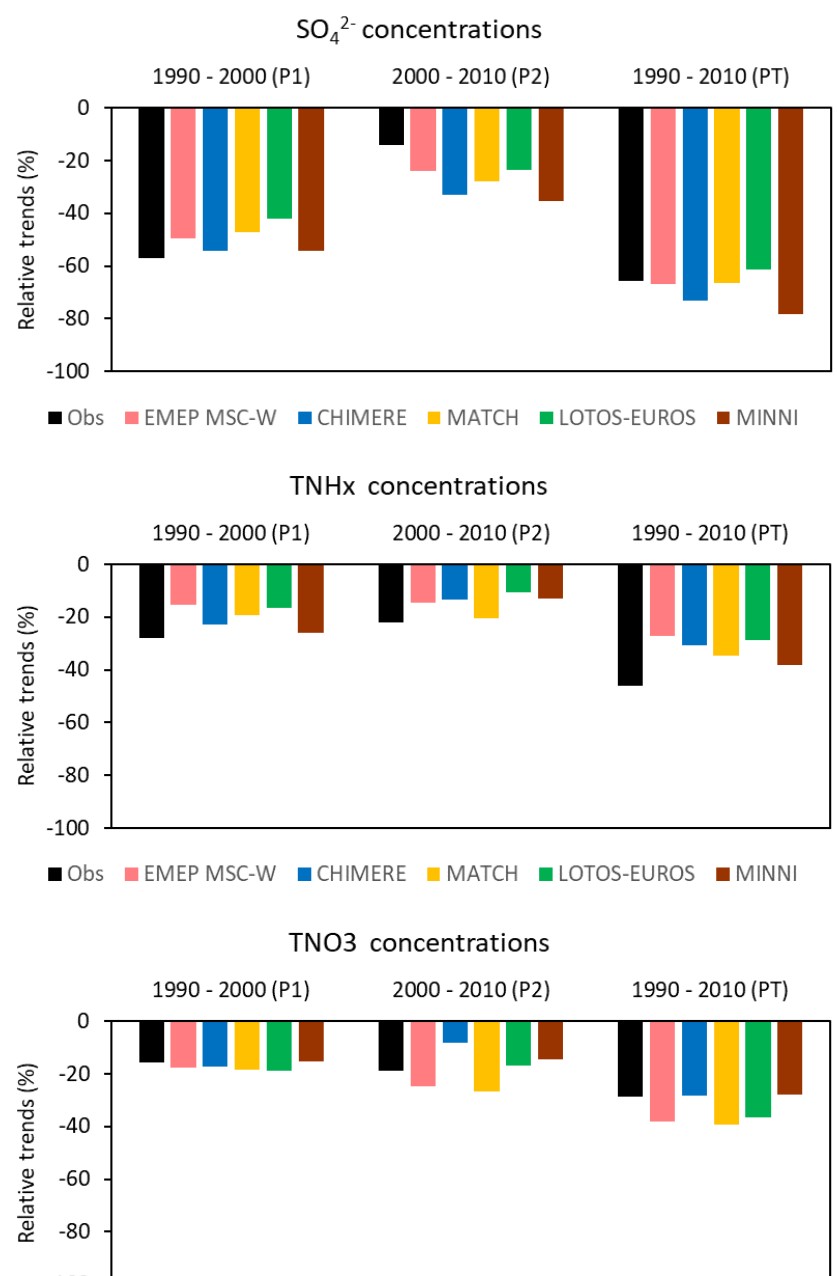

**Figure 6:** Modeled and observed (Obs) mean relative trends in $SO_4^{2-}$ (PSO4), $TNH_x$ and $TNO_3$ concentrations for the P1 (1990–2000), P2 (2000–2010) and PT (1990–2010) periods.





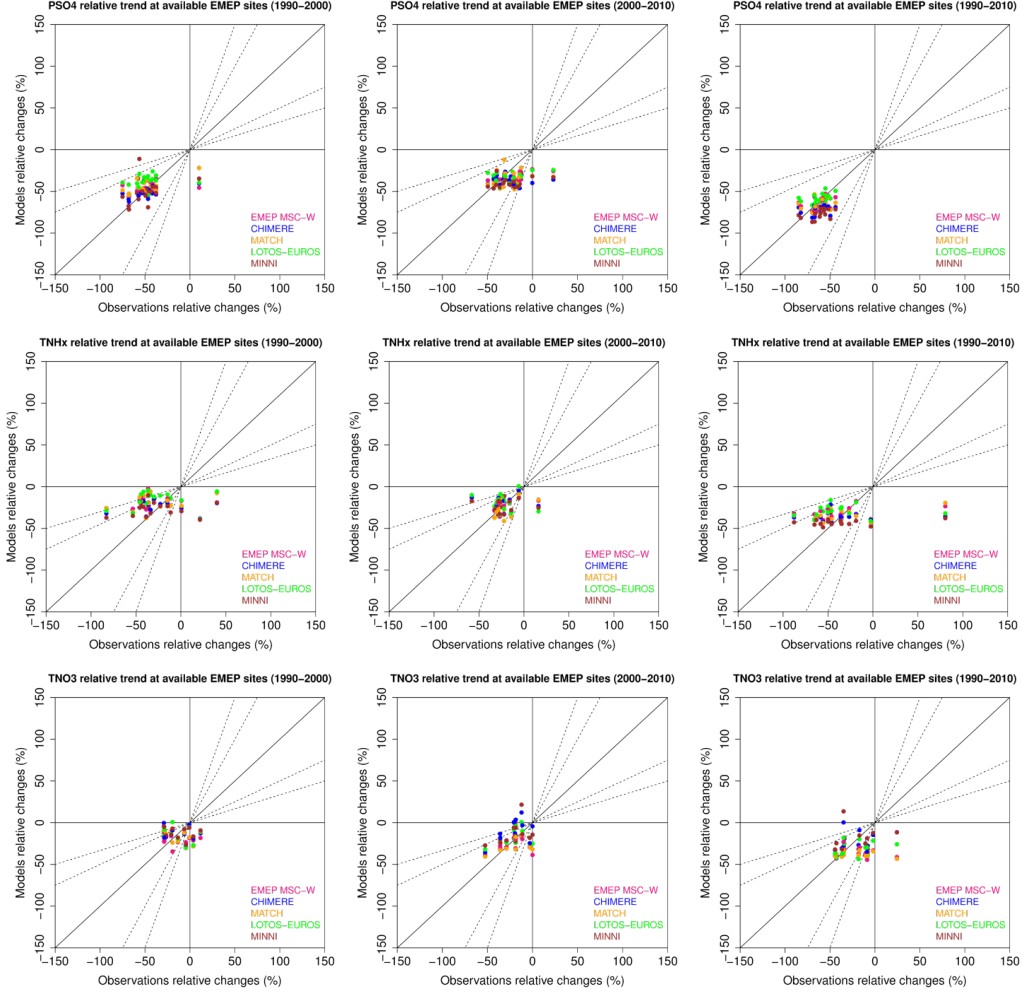

**Figure 7:** Modeled and observed $SO_4^{2-}$ (upper-panel), $TNH_x$ (mid-panel) and $TNO_3$ (lower-panel) relative trends for the P1 (1990–2000), P2 (2000–2010) and PT (1990–2010) periods (left to right). The continuous line indicates the 1:1 line, and the dotted lines indicate the 1:2 and 1:3 lines (and their reciprocals).







**Figure 8:** Percentage of statistically significant/non-significant increasing/decreasing trends in the observations and modeled data for SO₄²⁻ (top-panel) and TNHₓ (central-panel) and TNO₃ (bottom-panel) for the P1 (1990–2000), P2 (2000–2010) and PT (1990–2010) periods (from left to right).





**Figure 9:** Modeled HNO$_3$ and NO$_3^-$ relative trends over lands for the P1 (1990–2000, first and second columns) and P2 (2000–2010, third and second columns) periods as predicted by all the models (rows; from top to bottom: EMEP MSC-W, CHIMERE, MATCH, LOTOS-EUROS, MINNI). White areas indicate non-significant trends.





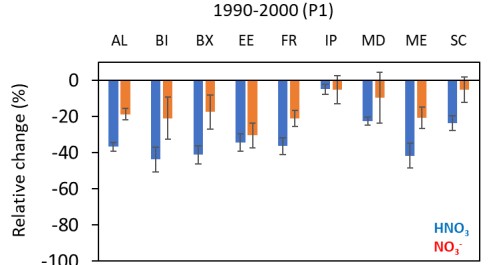
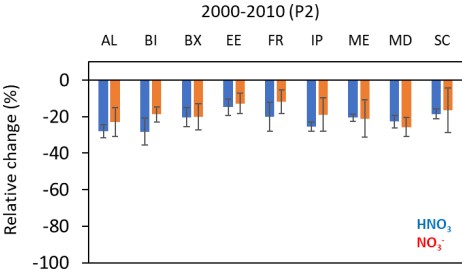

**Figure 10:** Modeled relative trends in HNO₃ and NO₃⁻ concentrations for the different PRUDENCE zones (Figure 1) for the P1 (left) and P2 (right) periods. The columns show the averages (over land) of all the model estimates and the bars show the standard deviation respect to models.





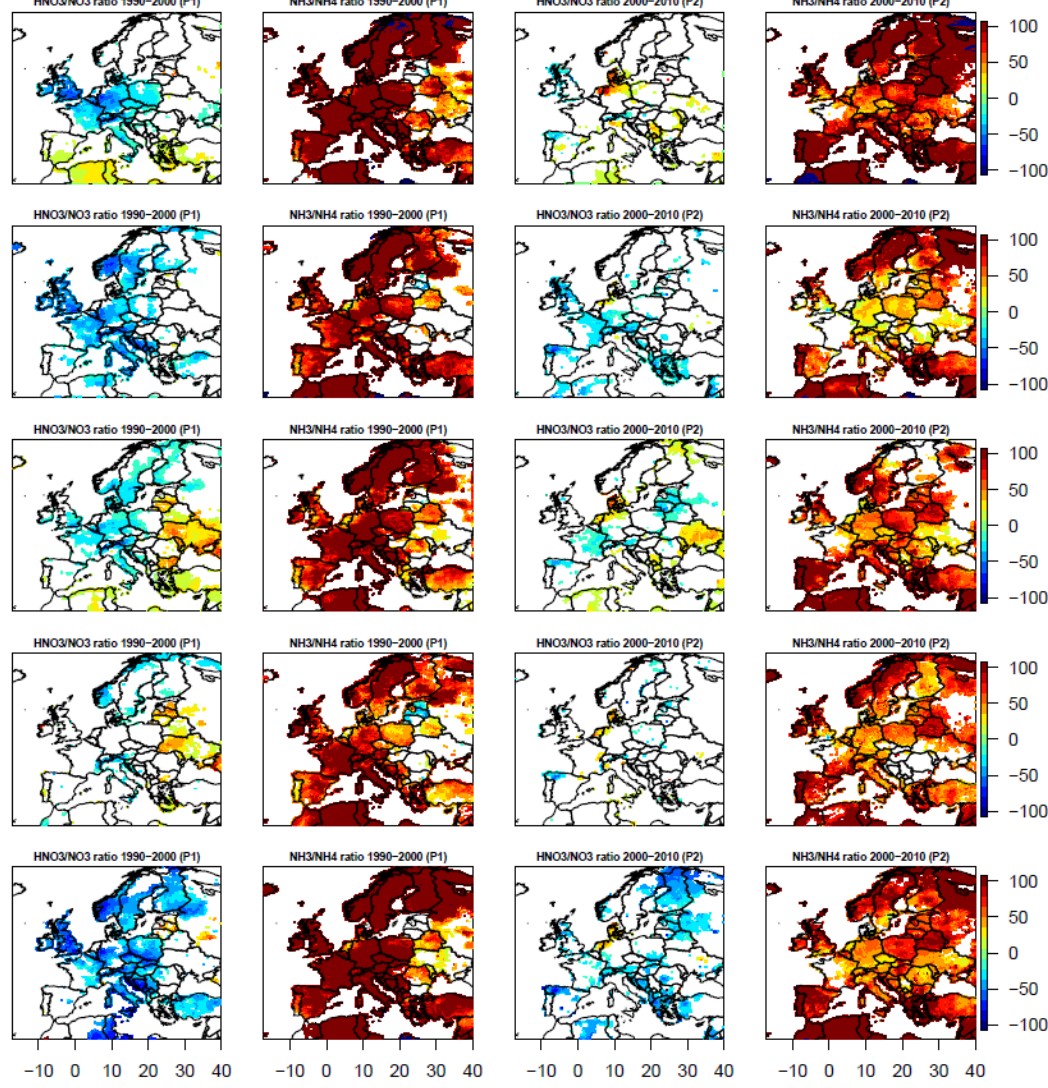

**Figure 11:** Modeled $HNO_3/NO_3^-$ and $NH_3/NH_4^+$ molar ratio relative trends over lands for the P1 (1990–2000, first and second columns) and P2 (2000–2010, third and second columns) periods as predicted by all the models (rows; from top to bottom: EMEP MSC-W, CHIMERE, MATCH, LOTOS-EUROS, MINNI). White areas indicate non-significant trends. Scale was saturate at 100% to facilitate the comprehension of the panel.





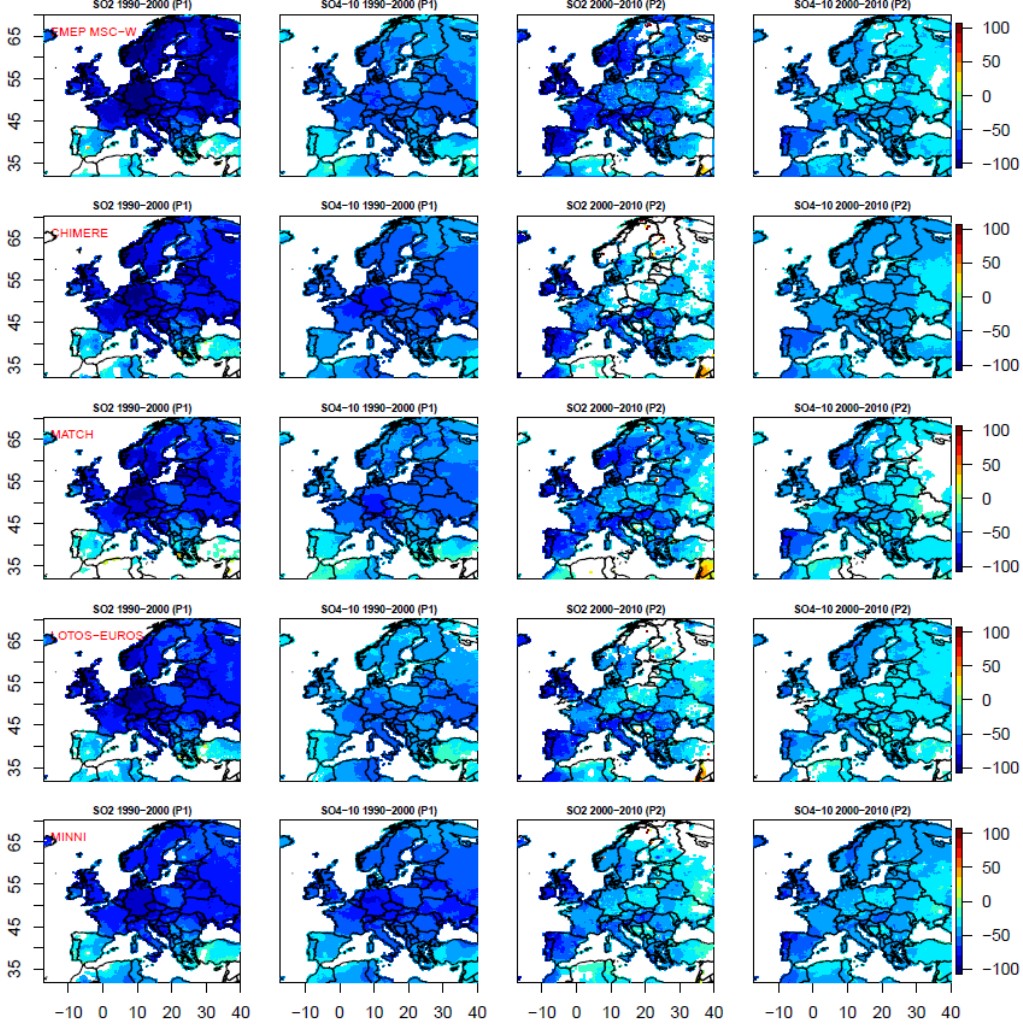

**Figure 12:** Modeled SO$_2$ and SO$_4^{2-}$ relative trends over lands for the P1 (1990–2000, first and second columns) and P2 (2000–2010, third and second columns) periods as predicted by all the models (rows; from top to bottom: EMEP MSC-W, CHIMERE, MATCH, LOTOS-EUROS, MINNI). White areas indicate non-significant trends. Scale was saturate at 100% to facilitate the comprehension of the panel.




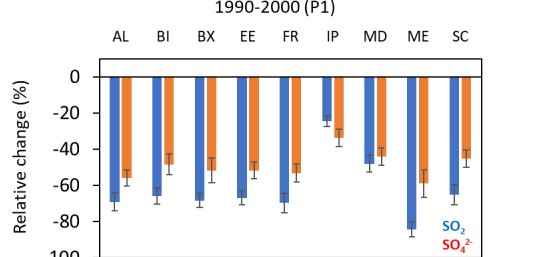
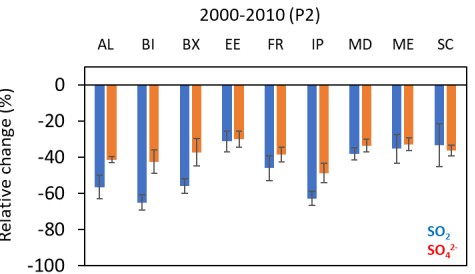

**Figure 13:** Modeled relative trends in $SO_2$ and $SO_4^{2-}$ concentrations for the different PRUDENCE zones (Figure 1) for the P1 (left) and P2 (right) periods. The columns show the averages (over land) of all the model estimates and the bars show the standard deviation respect to models.





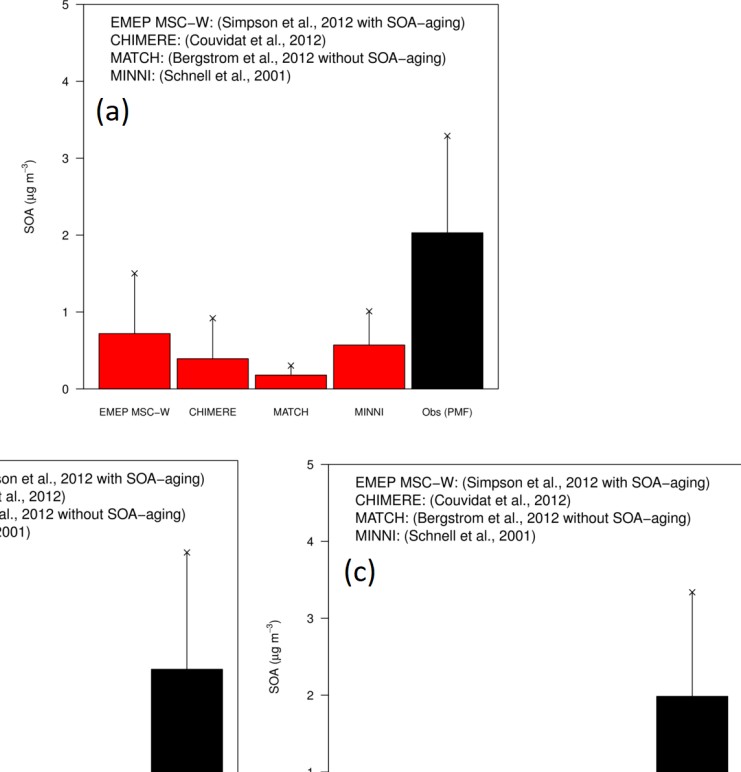

**Figure 14:** Modeled and observed (retrieved from PMF analysis) means and standard deviations of SOA concentrations (a) and for summer campaigns (b) and winter campaigns (c) (Table S2).







**Figure 15:** Modeled relative and absolute ASOA and BSOA fractions for summer (left-panels) and winter (right-panels) for different years (between 2000 and 2010) and seasons of the year (Table S2).





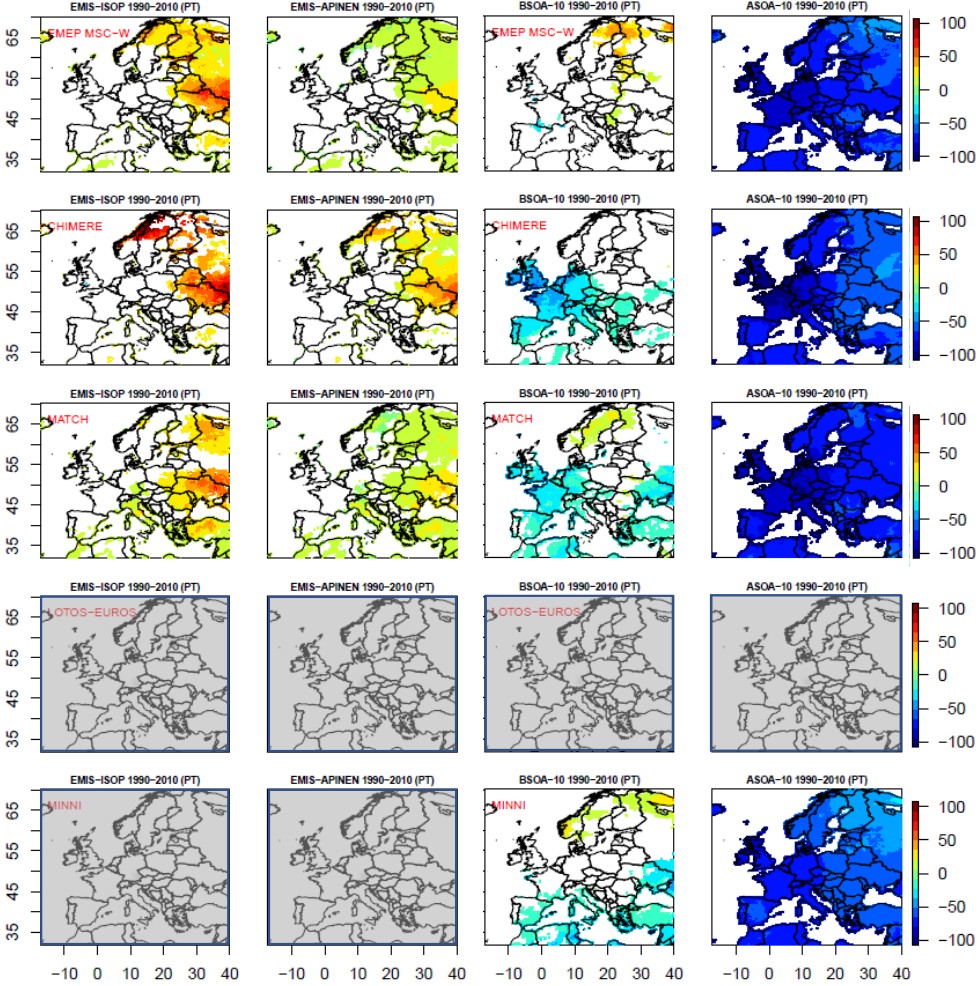

**Figure 16:** Modeled relative trends in isoprene and monoterpene emissions for the PT period (1990–2010, first and second column) and biogenic and anthropogenic SOA relative trends for the PT period (1990–2010, third and fourth column) as predicted by all the models (rows; from top to bottom: EMEP MSC-W, CHIMERE, MATCH, LOTOS-EUROS, MINNI). White areas indicate non-significant trends. Scale was saturate at 100% to facilitate the comprehension of the panel. Grey panels indicate missing data.





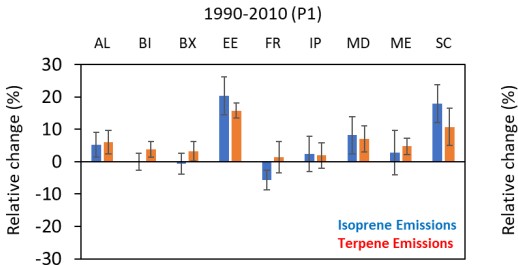
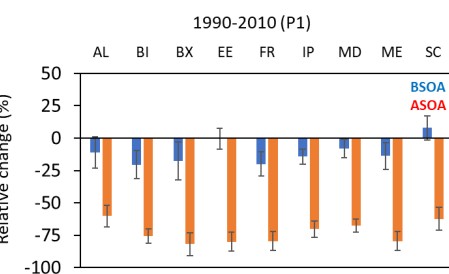

**Figure 17:** Modeled relative trends for the biogenic emissions (isoprene and terpene, left panel) and anthropogenic and biogenic SOA concentrations for the different PRUDENCE zones (Figure 1) for the 1990–2010 (PT) period. The columns show the averages (over land) of all the model estimates and the bars show the standard deviation respect to models.





**Table 1:** Chemical, thermodynamic schemes and biogenic emission models used by the modeling teams in the EURODELTA-Trends experiment.

| Model | Gas-phase chemistry | SIA Module | SOA module | VBS for aerosol | Biogenic model |
|---|---|---|---|---|---|
| CHIMERE (model version 2017β v1.0) | MELCHIOR2 (Derognat et al., 2003) | ISORROPIA v2.1 (Nenes et al., 1999) | H²O (Couvidat et al., 2012) coupled with SOAP (Couvidat and Sartelet, 2015) | Not used in this study | MEGAN v2.1 (Guenther et al., 2012) |
| EMEP MSC-W (model rv.4.7, spring 2015) | EmChem09 (Simpson et al., 2012) | MARS (Binkowski and Shankar, 1995) | VBS-NPAS (Simpson et al., 2012) | Yes (Bergström et al., 2012) | (Simpson et al., 2012) Based upon maps of 115 tree species from (Koeble and Seufert, 2001) |
| LOTOS-EUROS (model version 1.10.005) | TNO-CBM-IV (Schaap et al., 2009) | ISORROPIA II (Fountoukis and Nenes, 2007) | Not used in this study | Not used in this study | (Bergström et al., 2012) Based upon maps of 115 tree species from (Koeble and Seufert, 2001) |
| MATCH (model version April 2016) | Based on EMEP MSC-W (Simpson et al., 2012) with modified isoprene chemistry (Carter, 1996; Langner et al., 1998) | RH and T dependent equilibrium constant (Mozurkewich, 1993) | Similar to VBS-NPNA (Bergström et al., 2012) | Yes (Bergström et al., 2012) | (Bergström et al., 2012) Based upon maps of 115 tree species from (Koeble and Seufert, 2001) |
| MINNI (model version 4.7) | SAPRC99 (Carter, 2000) | ISORROPIA v1.7 (Nenes et al., 1998) | SORGAM module (Schell et al., 2001) | None | MEGAN v2.04 (Guenther et al., 2006) |





**Table 2:** Relative and absolute trends in emissions of SO$_x$, NO$_x$, NH$_3$ and NMVOCs in the EURODELTA-Trends exercise (whole domain). Trends are reported for the entire 1990–2010 period as well as for two sub-periods, 1990–2000 and 2000–2010. The linear trends were calculated using the Theil-Sen method (Sen, 1968).

| | 1990–2000 (P1) | | 2000–2010 (P2) | | 1990–2010 (PT) | |
|---|---|---|---|---|---|---|
| | Total relative change (%) | Absolute change per year (ktons y$^{-1}$) | Total relative change (%) | Absolute change per year (ktons y$^{-1}$) | Total relative change (%) | Absolute change per year (ktons y$^{-1}$) |
| SO$_x$ | -54 | -1 952 | -37 | -668 | -69 | -1 061 |
| NO$_x$ | -25 | -659 | -17 | -356 | -39 | -510 |
| NH$_3$ | -19 | -129 | -6 | -31 | -15 | -45 |
| NMVOCs | -33 | -812 | -33 | -525 | -59 | -705 |



**Table 3:** Modeled and observed mean relative trends of NO$_2$ and SO$_2$ for the P1 (1990–2000), P2 (2000–2010) and PT (1990–2010) periods and percentage of points in Figure 3 within a factor of 2 of the observed trends.

| NO$_2$ | P1 (%) | P2 (%) | PT (%) | P1 (% of points within a factor of 2) | P2 (% of points within a factor of 2) | PT (% of points within a factor of 2) |
|---|---|---|---|---|---|---|
| Obs | -25 | -12 | -36 | - | - | - |
| EMEP MSC-W | -19 | -22 | -44 | 56 | 56 | 52 |
| CHIMERE | -23 | -25 | -47 | 52 | 48 | 56 |
| MATCH | -20 | -26 | -46 | 64 | 52 | 52 |
| LOTOS-EUROS | -21 | -22 | -46 | 48 | 48 | 52 |
| MINNI | -19 | -24 | -44 | 52 | 56 | 56 |
| SO$_2$ | P1 (%) | P2 (%) | PT (%) | P1 (% of points within a factor of 2) | P2 (% of points within a factor of 2) | PT (% of points within a factor of 2) |
| Obs | -82 | -47 | -97 | - | - | - |
| EMEP MSC-W | -76 | -54 | -97 | 100 | 83 | 100 |
| CHIMERE | -69 | -34 | -91 | 97 | 63 | 100 |
| MATCH | -67 | -48 | -88 | 100 | 83 | 100 |
| LOTOS-EUROS | -69 | -40 | -88 | 97 | 67 | 100 |
| MINNI | -64 | -41 | -84 | 97 | 80 | 100 |





**Table 4:** Same as Table 3 but for $SO_4^{2-}$, $TNH_x$ and $TNO_3$.

| $SO_4^{2-}$ | P1 (%) | P2 (%) | PT (%) | P1 (% of points within a factor of 2) | P2 (% of points within a factor of 2) | PT (% of points within a factor of 2) |
|---|---|---|---|---|---|---|
| Obs | -57 | -14 | -66 | - | - | - |
| EMEP MSC-W | -49 | -24 | -67 | 95 | 65 | 100 |
| CHIMERE | -54 | -33 | -73 | 95 | 60 | 100 |
| MATCH | -47 | -28 | -67 | 95 | 65 | 100 |
| LOTOS-EUROS | -42 | -23 | -61 | 95 | 65 | 100 |
| MINNI | -54 | -35 | -78 | 90 | 65 | 100 |
| $TNH_x$ | P1 (%) | P2 (%) | PT (%) | P1 (% of points within a factor of 2) | P2 (% of points within a factor of 2) | PT (% of points within a factor of 2) |
| Obs | -28 | -22 | -46 | - | - | - |
| EMEP MSC-W | -15 | -14 | -27 | 44 | 75 | 75 |
| CHIMERE | -23 | -14 | -31 | 44 | 75 | 75 |
| MATCH | -19 | -21 | -35 | 44 | 81 | 81 |
| LOTOS-EUROS | -16 | -11 | -29 | 25 | 69 | 63 |
| MINNI | -26 | -13 | -38 | 56 | 81 | 75 |
| $TNO_3$ | P1 (%) | P2 (%) | PT (%) | P1 (% of points within a factor of 2) | P2 (% of points within a factor of 2) | PT (% of points within a factor of 2) |
| Obs | -16 | -19 | -29 | - | - | - |
| EMEP MSC-W | -18 | -25 | -38 | 62 | 85 | 54 |
| CHIMERE | -17 | -8 | -28 | 46 | 38 | 54 |
| MATCH | -18 | -27 | -39 | 46 | 77 | 54 |
| LOTOS-EUROS | -19 | -17 | -37 | 46 | 54 | 54 |
| MINNI | -16 | -14 | -28 | 31 | 54 | 54 |



**Table 5:** Means and standard deviations of the modelled relative changes in $HNO_3$ and $NO_3^-$ concentrations for the P1 (1990–2000) and P2 (2000–2010) periods for all the PRUDENCE regions (Figure 1).

| Regions | 1990–2000 (P1) | | 2000–2010 (P2) | |
|---|---|---|---|---|
| | $HNO_3$ Relative change (±SD) (%) | $NO_3^-$ Relative change (±SD) (%) | $HNO_3$ Relative change (±SD) (%) | $NO_3^-$ Relative change (±SD) (%) |
| AL | -37 (±2) | -19 (±3) | -28 (±4) | -23 (±8) |
| BI | -44 (±7) | -21 (±12) | -28 (±7) | -19 (±4) |
| BX | -41 (±5) | -18 (±9) | -20 (±5) | -20 (±7) |
| EE | -34 (±5) | -30 (±7) | -15 (±4) | -13 (±6) |
| FR | -36 (±5) | -21 (±4) | -20 (±8) | -12 (±7) |
| IP | -5 (±3) | -5 (±8) | -26 (±3) | -19 (±9) |
| MD | -23 (±2) | -10 (±14) | -21 (±2) | -21 (±10) |
| ME | -42 (±7) | -21(±6) | -23 (±3) | -26 (±5) |
| SC | -24 (±4) | -5 (±7) | -19 (±3) | -16 (±12) |

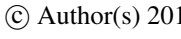



**Table 6:** Means and standard deviations of the modelled relative changes in $SO_2$ and $SO_4^{2-}$ concentrations for the P1 (1990–2000) and P2 (2000–2010) periods for all the PRUDENCE regions (Figure 1).

| Regions | 1990–2000 (P1) | | 2000–2010 (P2) | |
| --- | --- | --- | --- | --- |
| | $SO_2$ Relative change (±SD) (%) | $SO_4^{2-}$ Relative change (±SD) (%) | $SO_2$ Relative change (±SD) (%) | $SO_4^{2-}$ Relative change (±SD) (%) |
| AL | -69 (±5) | -56 (±4) | -57 (±7) | -41 (±2) |
| BI | -66 (±4) | -49 (±6) | -65 (±4) | -43 (±6) |
| BX | -69 (±4) | -52(±9) | -56 (±4) | -37 (±8) |
| EE | -67 (±4) | -52 (±5) | -31 (±6) | -30 (±4) |
| FR | -70 (±6) | -53 (±5) | -46 (±7) | -39 (±4) |
| IP | -24 (±3) | -34 (±5) | -63 (±4) | -49 (±5) |
| MD | -48 (±5) | -44 (±5) | -38 (±3) | -34 (±3) |
| ME | -85 (±4) | -59 (±8) | -35 (±8) | -33 (±4) |
| SC | -65 (±6) | -45 (±5) | -33 (±12) | -36 (±3) |





**Table 7:** Observed and predicted SOA concentrations, averaged over all sites (Table S2 and Figure S1), and for the summer and winter campaigns. Statistics are normalized mean bias (NMB), normalized mean error (NME), Mean Bias (MB) and mean absolute gross error (MAGE).

| | | Mean Predicted ($\mu g\ m^{-3}$) | Mean Observed ($\mu g\ m^{-3}$) | NMB (%) | NME (%) | MB ($\mu g\ m^{-3}$) | MAGE ($\mu g\ m^{-3}$) |
|---|---|---|---|---|---|---|---|
| Winter | EMEP MSC-W | 0.21 | 1.98 | -0.90 | 0.90 | -1.78 | 1.78 |
| | CHIMERE | 0.12 | 1.98 | -0.94 | 0.94 | -1.87 | 1.87 |
| | MATCH | 0.12 | 1.98 | -0.94 | 0.94 | -1.86 | 1.86 |
| | MINNI | 0.29 | 1.98 | -0.85 | 0.85 | -1.69 | 1.69 |
| Summer | EMEP MSC-W | 1.42 | 2.29 | -0.38 | 0.44 | -0.87 | 1.00 |
| | CHIMERE | 0.85 | 2.29 | -0.63 | 0.63 | -1.43 | 1.43 |
| | MATCH | 0.25 | 2.29 | -0.89 | 0.89 | -2.04 | 2.04 |
| | MINNI | 0.52 | 2.29 | -0.77 | 0.77 | -1.76 | 1.76 |
| All Periods | EMEP MSC-W | 0.72 | 2.03 | -0.65 | 0.73 | -1.31 | 1.48 |
| | CHIMERE | 0.39 | 2.03 | -0.81 | 0.84 | -1.64 | 1.70 |
| | MATCH | 0.18 | 2.03 | -0.91 | 0.91 | -1.85 | 1.85 |
| | MINNI | 0.57 | 2.03 | -0.72 | 0.79 | -1.46 | 1.60 |