# Peer review of "Trends of inorganic and organic aerosols and precursor gases in Europe: insights from the EURODELTA multi-model experiment over the 1990–2010 period"

_Geoscientific Model Development, 2019_

## Referee Comment (RC1) · Anonymous Referee #1 · 1 May 2019

**General and Specific Comments**

The manuscript describes the trends in simulated atmospheric concentration of several gas and particulate pollutants across Europe and compares them to observed concentration at long-term monitoring EMEP sites, along with emissions of major species. Simulated concentrations proceed from 5 different Chemical Transport Models within the joint effort EURODELTA-Trends, whose results and main description was previously presented by Colette et al. (2017).

[Figure]

The study aims to reply to a question of main interest for the scientific community, i.e. the main drivers of the variation in atmospheric composition over the two decades 1990 – 2010. The paper is clearly written, the material is well presented and most of the analysis is solid. However some comments and margin of improvement can be explored.

1. Unless the authors would identify suitable ground stations in regions IP, FR, MD for the comparison of the trends in these large areas of the continent, a comment about the limited number of sites for trend comparison would be needed: for some species (e.g. TNO3), more than half of the continent could not provide any data, constraining the spatial validity of the results presented.

2. The trend analysis is extremely basic. Several methods, which are often ready-to-use in several software packages, can provide the uncertainty bands for these trends and a better estimate of their significance. It is not clear whether the trend analysis was applied on daily, monthly or annual data, although the slope and the significance of a trend is highly dependent on the autocorrelation of the data, mainly for Theil-Sen's and the Mann-Kendall methods (see Collaud-Coen, 2013).

3. There is no indication about the performance of these CTMs, neither in this manuscript nor in Colette et al. (2017), notwithstanding the several indications in the scientific literature or by the FAIRMODE initiative (https://fairmode.jrc.ec.europa.eu/). A brief summary of the "quality" of the simulation output should be included, which is something different, but complementary, from the scatter plots for trends.

4. The data available to the authors through EURODELTA-Trends could be explored to understand why in some circumstances one scheme is better than another: part

of this analysis has been performed for the Secondary Organic Aerosol (SOA). There is a potential to perform a similar analysis also for the species presenting the largest variability among the models (e.g. TNO3 over P2): some hints and guidance about this point could be useful for the modelling community.

5. In page 11, line 12 the authors suggest that the simulated results for NO2 could be driven by an "overestimated negative trends in national emission data bases". There is no indication about the version of the COPERT model used for the emission factor of transportation, although the last part of the investigated period is included in the "emission scandal". This point should be considered.

**Minor comments**

- p. 5, l . 33: wrong parenthesis "can be found in Colette et al. (2017)"

- p. 9, l. 38: why exactly 7 $\mu$g/m$^3$ was chosen? Please explain

- p.11 sub-paragraph 3.2.1: Please organize it in order along with the title "...SO2 and NO2 concentration trends": therefore it should be organized with relative trends for SO2 first and then relative trends for NO2. This is already done for the second part of the sub-paragraph about the significance.

- p.12, l. 15: full stop is missing "... influenced by different factors. First ..."

- p.15 l. 21: ASOA and BSOA acronym were defined only in the abstract and not in the main text

- p. 16 l. 3: misspelling terpene as singular? "i.e. isoprene and terpenes"

- p. 16 l. 18: misspelling of plays "could also play"

- p. 17 l. 37: maybe you meant "latter" instead of "later"

- p. 18 l. 13: maybe you meant "monoterpenes"?

- p. 18 l. 14: this is the first comment about the influence of the change in Land Use, it is better to mention this in the discussion paragraph

- Range of y-axis for figure 14 and 15 maybe set to 4 $\mu$g/m$^3$, actually figure 15 could be reduced to 3 $\mu$g/m$^3$: this would slightly improve its readability.

**References**

Colette, A., Andersson, C., Manders, A., Mar, K., Mircea, M., Pay, M.-T., Raffort, V., Tsyro, S., Cuvelier, C., Adani, M., Bessagnet, B., Bergström, R., Briganti, G., Butler, T., Cappelletti, A., Couvidat, F., D'Isidoro, M., Doumbia, T., Fagerli, H., Granier, C., Heyes, C., Klimont, Z., Ojha, N., Otero, N., Schaap, M., Sindelarova, K., Stegehuis, A. I., Roustan, Y., Vautard, R., van Meijgaard, E., Vivanco, M. G., and Wind, P.: EURODELTA-Trends, a multi-model experiment of air quality hindcast in Europe over 1990–2010, Geosci. Model Dev., 10, 3255-3276, https://doi.org/10.5194/gmd-10-3255-2017, 2017.

Collaud Coen, M., Andrews, E., Asmi, A., Baltensperger, U., Bukowiecki, N., Day, D., Fiebig, M., Fjaeraa, A. M., Flentje, H., Hyvärinen, A., Jefferson, A., Jennings, S. G., Kouvarakis, G., Lihavainen, H., Lund Myhre, C., Malm, W. C., Mihapopoulos, N., Molenar, J. V., O'Dowd, C., Ogren, J. A., Schichtel, B. A., Sheridan, P., Virkkula, A., Weingartner, E., Weller, R., and Laj, P.: Aerosol decadal trends – Part 1: In-situ optical measurements at GAW and IMPROVE stations, Atmos. Chem. Phys., 13, 869-894, https://doi.org/10.5194/acp-13-869-2013, 2013.

---

## Referee Comment (RC2) · Anonymous Referee #2 · 17 Jun 2019

**Review comments to manuscript gmd-2019-70:**

**Trends of inorganic and organic aerosols and precursor gases in Europe: insights from the EURODELTA multi-model experiment over the 1990–2010 period**

**by Ciarelli et al.**

**General Comments**

The manuscript investigates trends in aerosol concentrations and chemical composition as well as their precursors in Europe over two decades that have experienced significant changes in emissions. The analysis is based on output of a suite of air quality models and observational data. The paper is well written and figures and tables are clear and informative. The paper contributes to the field of models evaluation and sensitivity analysis to different parameterizations and settings and will be of interest to GMD readers. However the following major points and specific comments could be addressed to improve it.

- The EURODELTA experiment includes output from eight air quality models of which five are included in the current manuscript. Although the additional three models do not allow a proper quantification of the trends they could be included as a model intercomparison exercise. Those models may indeed adopt more sophisticated (or computationally demanding) settings whose skills need to be quantified to better inform model users on the optimal model setup (see more details comments below). I would suggest including a new model intercomparison and evaluation section and present multiple statistical metrics of model skill for the years when all eight models have data.
- Despite the paper focuses on the quantification of temporal trends, it is important to quantify the actual model skills vs observations. This could be useful also in the context of informing users on the optimal model setup to be adopted when aiming to simulate aerosol concentration and composition in future studies. It would be also useful to add results of these analyses to the conclusions, where the model/settings with highest skills are highlighted.
- The trend analysis are performed for different aerosol species and over different regions. However there is no clear quantification of the spatial variability of these trends and how the model performance varies by year. It could be interesting to expand the current analyses and include a more detailed investigation on changes of model performance over different years and different sites. This will also contribute to a better quantification of inter-model variability and identification of sources of uncertainty in model output.

**Specific Comments**

- Despite there is no word limit for GMD abstracts, the current one is very long, so I would suggest to present the results in a more concise way (e.g. by grouping models/species having similar performance/behavior, etc.).
- Page 2, line 7: I would use "evaluate" instead of "validate" here and in other instances in the paper since you are performing a model evaluation (not validation).

- Page 2, line 13: the meaning of relative trend or relative reduction (here) was never defined in the paper. It should be clarified here and/or in the methods since this terminology is used in many instances in the text ad figures.
- Page 4, line 16: the years of the AeroCom experiment could be included
- Page 5, line 24: what is the spatial resolution of the EDT models? Is it the same for all of them? This information should be included.
- Page 7, line 36: LAI from MODIS is available only from 2002. What did the EDT simulations use for prior years?
- Page 8, line 15: it would be interesting to have some more details about the temporal variability of emissions since they are one of the key drivers of changes in aerosol properties. This would also support your statement that trends in emissions are non-linear and would be helpful to interpret the results at finer temporal scales. Some information about the how trends in emissions change in space would be also relevant (you could also refer to literature studies if available).
- Page 9, line 27: this claim could be supported by a supplementary figure or referring to the literature, as suggested in a prior comment.
- Page 10, line 5-16: these claims are not supported by your analyses. Could you show some more analyses and explain how the trends vary in space (or refer to the literature)?
- Page 11, line 20: Period is missing at the end of the sentence. Also it is not clear the number of stations, data availability and frequency of the data used for the analysis.
- Page 12, line 6: is there a way for you to indicate which dots these sites correspond to in Figure 7?
- Page 13, line 6: do you have observations to be used for model evaluation in Figure 9/10?
- Page 13, line 24: this claim could be better supported by referring to the spatial patterns of emissions, as suggested to do in previous comments.

**Tables**

Table 7: there is no mention of Table 7 in the manuscript but I would keep it, so please add reference to it and comments. Also the statistical metrics should be defined. What is the temporal resolution/frequency of the data used in this evaluation and how many observations do you have at each site?

**Figures**

Figure 5 and 8: not clear how many observational sites are included and the temporal resolution of the data used.

Figure 14: Rephrase caption to clarify that panel a includes all seasons as in Table S2, while summer and winter campaigns are shown in panels b and c.

Figure S1: why averages above 7 $\mu$g m$^{-3}$ are excluded? Explain this also in Table S1.

**Technical corrections:**

- Page 2, line 34: be consistent between "emission" and "emissions".
- Page 3, line 10: remove "the formation of".

- Page 4, line 29: SIA was not defined earlier.
- Page 8, line 35: typo "Weather Research and Forecasting".
- Page 9, line 11: typo "illustrates".
- Page 12, line 24: replace "model" with "models".
- Page 17, line 5: add also spatial resolution (if it is the same).
- Page 17, line 21: $SO_x$ missing subscript.
- Figure 2: Missing subscripts on titles for chemical species.

---

## Short Comment (SC1) · 2 Jul 2019

The code and data availability section in this manuscript fails to comply with GMD requirements for transparency and archiving in a number of ways. These need to be remedied before the manuscript can be accepted.

**1 EMEP data**

The documentation cited here is on a wiki server. This is an entirely ephemeral location which could change at any time. It is important that the information required to replicate the data are persistently available. The website pointed to is quite short, so one way to achieve this would be to reformat the content as a short report which could be archived in one of the following ways:

1. As an appendix to the paper.

2. On a preprint server such as arXiv.

3. On a persistent data repository such as https://zenodo.org

Care needs to be taken that this information is comprehensive (i.e. everything needed to rerun the experiment has been specified) and that the references to data stored on the THREDDS server are sufficiently precise that the exact data can be retrieved. Ideally this would be by attaching persistent identifiers such as DOIs to the data and referring to these.

The THREDDS server also seems to be inaccessible. Are these really the correct citations?

**2 Results data**

The results data section just links to a wiki page which is mostly about how to use ssh. No mechanism is provided for identifying the results data. Please ensure that the location of the results data is sufficiently precisely identifed that the reader could trace back from the results presented in the paper to the data from which those results were

computed. Once again, please ensure that this information is in a persistent archived location (in particular not a wiki page or other website) so that there can be some confidence that future readers will be able to access this information.

**3  Evaluation code**

Providing code on request does not satisfy GMD's requirements for transparency and reproducibility. The reasons for this are given in detail in the new editorial https://doi.org/10.5194/gmd-12-2215-2019. Please therefore provide a persistent, public archive of the evaluation code, for example on https://zenodo.org.

---

## Author Comment (AC1) · 11 Sep 2019

**Responses to the comments of anonymous referee #1**

Thank you for your comments that helped to improve our manuscript. Please find below your comments in black, our responses in blue and modifications in the revised manuscript in *italic*.

The manuscript describes the trends in simulated atmospheric concentration of several gas and particulate pollutants across Europe and compares them to observed concentration at long-term monitoring EMEP sites, along with emissions of major species. Simulated concentrations proceed from 5 different Chemical Transport Models within the joint effort EURODELTA-Trends, whose results and main description was previously presented by Colette et al. (2017). The study aims to reply to a question of main interest for the scientific community, i.e. the main drivers of the variation in atmospheric composition over the two decades 1990 – 2010. The paper is clearly written, the material is well presented and most of the analysis is solid. However some comments and margin of improvement can be explored.

We thank the referee for the comments on the manuscript.

1. Unless the authors would identify suitable ground stations in regions IP, FR, MD for the comparison of the trends in these large areas of the continent, a comment about the limited number of sites for trend comparison would be needed: for some species (e.g. TNO3), more than half of the continent could not provide any data, constraining the spatial validity of the results presented.

We agree with the referee and we added an additional comment on the spatial distribution and availability of the measurements data at line 13 of page 9 of the revisited manuscript as shown below:

*"It can be noted that most of the stations are located over the Northern and Central part of the domain, therefore limiting the evaluation of the model results to these specific sites".*

2. The trend analysis is extremely basic. Several methods, which are often ready-to-use in several software packages, can provide the uncertainty bands for these trends and a better estimate of their significance. It is not clear whether the trend analysis was applied on daily, monthly or annual data, although the slope and the significance of a trend is highly dependent on the autocorrelation of the data, mainly for Theil-Sen's and the Mann-Kendall methods (see Collaud-Coen, 2013).

We thank the referee for this comment. The trend analysis was applied to the annual observational datasets. We added this information at line 3 of page 9 of the revisited manuscript as below:

*"The annual observational datasets, [...]"*

We agree with the referee and we performed an additional trend analysis using a generalized least squares (GLS) fit model that accounts for the temporal autocorrelation of the data (using the "nlme" package available for R). The results from the GLS model are in line with the one predicted by the Theil-Sen's and the Mann-Kendall methods. For $NO_2$ and $SO_2$ gas-phase precursors, the GLS model showed a slightly higher fraction of non-significant decreasing trends during the P1, P2 and PT periods compared to the current analysis (Figure S2).

[Figure]

Figure S2: Percentage of statistically significant/non-significant (Si, Ns) increasing/decreasing (In, De) trends in the observations and modeled data for SO$_2$ (upper-panel) and NO$_2$ (lower-panel) for the P1 (1990–2000), P2 (2000–2010) and PT (1990–2010) periods (left to right) using a generalized least squares (GLS) fit model.

For SO$_4^{2-}$ and TNO$_3$, a higher fraction of non-significant increasing trends during the P2 period in the observational data was retrieved with the GLS fit model, whereas for TNH$_x$ a higher fraction of non-significant decreasing trends is predicted in the observational data during the P1 period compared to the Theil-Sen's and the Mann-Kendall methods (Figure S3).

[Figure]

Figure S3: Percentage of statistically significant/non-significant (Si, Ns) increasing/decreasing (In, De) trends in the observations and modeled data for $SO_4^{2-}$ (upper panel), $TNH_x$ (middle panel) and $TNO_3$ (lower panel) for the P1 (1990–2000), P2 (2000–2010) and PT (1990–2010) periods (left to right) using a generalized least squares (GLS) fit model.

In addition, the relative trends of $SO_2$, $SO_4^{2-}$, $NO_2$, $TNO_3$ and $TNH_x$, for all models and periods were also in line to the one predicted with the Theil-Sen's and the Mann-Kendall methods, with only small differences among the species and models (Table S3, below).

Table S3: Modeled and observed mean relative trends of $SO_2$, $SO_4^{2-}$, $NO_2$, $TNO_3$ and $TNH_x$ for the P1 (1990–2000), P2 (2000–2010) and PT (1990–2010) periods using a generalized least squares (GLS) fit model.

| | | $SO_2$ | $SO_4^{2-}$ | $NO_2$ | $TNO_3$ | $TNH_x$ |
|---|---|---|---|---|---|---|
| | Obs | -84 | -57 | -24 | -17 | -39 |
| | EMEP MSC-W | -77 | -48 | -17 | -20 | -19 |
| P1 | CHIMERE | -70 | -53 | -20 | -16 | -23 |
| | MATCH | -67 | -49 | -19 | -15 | -22 |
| | LOTOS-EUROS | -70 | -42 | -17 | -18 | -18 |
| | MINNI | -66 | -54 | -17 | -14 | -28 |
| | Obs | -42 | -13 | -13 | -23 | -22 |
| | EMEP MSC-W | -54 | -23 | -15 | -25 | -15 |
| P2 | CHIMERE | -22 | -31 | -21 | -9 | -12 |
| | MATCH | -46 | -29 | -22 | -26 | -16 |
| | LOTOS-EUROS | -35 | -29 | -20 | -16 | -11 |
| | MINNI | -39 | -34 | -15 | -16 | -16 |
| | Obs | -84 | -66 | -36 | -26 | -52 |
| | EMEP MSC-W | -86 | -66 | -40 | -37 | -29 |
| PT | CHIMERE | -75 | -71 | -45 | -26 | -31 |
| | MATCH | -80 | -68 | -44 | -37 | -34 |
| | LOTOS-EUROS | -77 | -61 | -41 | -34 | -28 |
| | MINNI | -75 | -75 | -39 | -27 | -39 |

We added the following paragraph at line 40 of page 11 of the revisited manuscript as well as Figure S2 and Table S3 in the revisited supplementary material:

"*An additional trend analysis was performed using a Generalized Least Squares (GLS) fit model that accounts for the temporal autocorrelation of the data. The results from the GLS model were in line with the one predicted by the Theil-Sen's and the Mann-Kendall methods (Figure S3 and Table S3), and* with *the GLS model showing a slightly higher fraction of non-significant decreasing trends during the P1, P2 and PT periods*".

Also, we added the following paragraph at line 15 of page 13 of the revisited manuscript as well as Figure S3 in the revisited supplementary material:

"*As for the $SO_2$ and $NO_2$ gas-phase species, an additional trend analysis was performed using a GLS fit model. For $SO_4^{2-}$, $TNO_3$ and $TNH_x$ results were also in line with the one predicted by the Theil-Sen's and the Mann-Kendall methods (Figure S3 and Table S3). For $SO_4^{2-}$ and $TNO_3$, the GLS model showed higher fraction of non-significant increasing trends during the P2 period in the observations data, whereas for $TNH_x$ an higher fraction of non-significant decreasing trends was retrieved compared to the Theil-Sen's and the Mann-Kendall methods during the P1 period*".

We would like to notice that Figure 5 and 8 of the revisited manuscript have been updated correct for a bug issue identified through careful additional visual analysis of the data. The conclusions remained unchanged apart for the $SO_4^{2-}$ during the P2 periods, where the models tend to over-predict the fraction of statistically significant decreasing trends, and for $TNO_3$ during the PT periods, where a higher percentage of non-significant decreasing trends are predicted (both in measurements and model data). Small changes also occurred for $NO_2$ during the P2 period where a higher percentage of non-significant decreasing trends is predicted (both in measurements and model data) and $TNH_x$ with the models that still tend to over-predict the number of statistically significant decreases. We updated the following sentences at line 22 of page 12 of the revisited manuscript as below:

*"The percentage of statistically significant/non-significant increasing/decreasing trends in the observed and modeled $SO_4^{2-}$ trends is reported in Figure 8, showing a good agreement between the observed and modeled significances (and their direction) for the P1 and PT periods, whereas all the models tend to over-predict the number of statistically significant increasing trends during the P2 period."*

At line 20 of page 2 in the abstract section as below:

*"even though all the models over-predicted the number of statistically significant decreasing trends during the P2 period".*

At line 33 of page 11 as below:

*"was only partially reproduced by the models".*

At line 38 of page 11 as below:

*" (the significant increase in the LOTOS-EUROS model is for the SE0014R station located in Sweden)".*

As well as at line 2 of page 13 as below:

*"during the P2 period".*

3. There is no indication about the performance of these CTMs, neither in this manuscript nor in Colette et al. (2017), notwithstanding the several indications in the scientific literature or by the FAIRMODE initiative (https://fairmode.jrc.ec.europa.eu/). A brief summary of the "quality" of the simulation output should be included, which is something different, but complementary, from the scatter plots for trends.

We agree with the referee and we have included a summary of the quality of the simulations for $NO_2$, $SO_2$, $SO_4^{2-}$, $TNO_3$ and $TNH_x$ as also suggested by the second referee. We calculated the mean fractional bias (MFB) and mean fractional errors (MFE) for all the models and species available in the exercise and for the two different periods (P1 and P2). The results were compared with the model performance criteria and performance goal proposed by Boylan and Russell, 2006.
Results indicated that the model performance criteria (MFB ≤ ±60 %, MFE ≤ +75 %) was met for all the species and by all the models that participated in the EURODELTA-Trends exercise, whereas the performance goal (MFB ≤ ±30 %, MFE ≤ +50 %) differs for some of the models and species. For instance, $TNO_3$ goal performance was not met by EMEP MSC-W in the P1 period and by CHIMERE for the P1 and P2 periods. Also, $SO_2$ performance goal was not met by MATCH during the P2 period and by MINNI in both the P1 and P2 period (Figure S4, below).

[Figure]

[Figure]

Figure S4: Soccer-goal plots for yearly concentrations of $SO_2$, $SO_4^{2-}$, $NO_2$, $TNO_3$ and $TNH_x$ for the 1990–2000 (P1) period (left) and 2000–2010 (P2) period (right). Rows; from top to bottom: EMEP MSC-W, CHIMERE, MATCH, LOTOS-EUROS, MINNI. MFB: mean fractional bias; MFE: mean fractional error.

We added the following paragraph at line 1 of page 12 of the revisited manuscript and as well as Figure S4 on the revisited supplementary material:

*"Model performance for $SO_2$ and $NO_2$ was additionally evaluated by calculating the mean fractional bias (MFB) and mean fractional error (MFE) for both the P1 and P2 periods separately (Appendix A). Recommended model performance criteria (MFB ≤ ±60 %,MFE ≤ +75 %) as well as the performance goal (MFB ≤ ±30 %, MFE ≤ +50 %) proposed by Boylan and Russell, 2006 were achieved in both periods by most of the models apart for $SO_2$ for MINNI during the P1 and P2 period and for MATCH during the P2 period where only the model performance criteria was achieved (Figure S4)."*

and the following paragraph at line 20 of page 13 of the revisited manuscript:

*"Model performance for $SO_4^{2-}$, $TNO_3$ and $TNH_x$ was also satisfactory; the recommended model performance criteria (MFB ≤ ±60 %,MFE ≤ +75 %) as well as the performance goal (MFB ≤ ±30 %, MFE ≤ +50 %) proposed by Boylan and Russell, 2006 were achieved in both the P1 and P2 periods by most of the models apart for $TNO_3$ in CHIMERE during the P1 and P2 periods and in EMEP MSC-W during the P1 period where only the model performance criteria were achieved (Figure S4)."*

4. The data available to the authors through EURODELTA-Trends could be explored to understand why in some circumstances one scheme is better than another: part of this analysis has been performed for the Secondary Organic Aerosol (SOA). There is a potential to perform a similar analysis also for the species presenting the largest variability among the models (e.g. TNO3 over P2): some hints and guidance about this point could be useful for the modelling community.

We recognize the importance of the different chemical schemes used in the models, however, for the purpose of this study we preferred to focus on the capability of the models in reproducing the observed trends rather than performing a specific model-to-model evaluation of the single schemes. Nevertheless, we included a summary of the quality of the single model performance for $SO_2$, $SO_4^{2-}$, $NO_2$, $TNO_3$ and $TNH_x$ for the two sub-periods (P1 and P2) as presented in the previous comment.
We additionally performed an evaluation of the evolution of the models mean fraction bias over the full 1990–2010 period for all the species and models that participated in the EURODELTA-Trends exercise. Results indicated that the mean fractional bias did not change substantially over the investigated period apart from $SO_2$ and $TNH_x$ where most of the models tend to be slightly more positively biased in the latter part of the period compared to 1990 (Figure S5, below).

[Figure]

[Figure]

[Figure]

[Figure]

[Figure]

Figure S5: Evolution of the mean fractional bias (MFB) for $SO_2$, $SO_4^{2-}$, $NO_2$, $TNO_3$ and $TNH_x$ over the 1990–2010 period.

We added the following paragraph at line 6 of page 12 of the revisited manuscript and as well as Figure S5 in the revisited supplementary material.

*"In addition, the evolution of the MFB over the full 1990–2010 period does not indicate any substantial change in 2010 compared to the first year of the exercise (i.e. 1990) with the exception of $SO_2$ which tends to be slightly more positively biased in the latter part of the period compared to 1990 (apart from EMEP MCS-W, Figure S5)."*

and the following paragraphs at line 24 of page 13 of the revisited manuscript.

*In addition, the evolution of the MFB over the full 1990–2010 period does not indicate any substantial change in 2010 compared to the first year of the exercise (i.e. 1990) apart from $TNH_x$ which tends to be slightly more positively biased in the latter part of the period compared to 1990 (Figure S5).*

5. In page 11, line 12 the authors suggest that the simulated results for NO2 could be driven by an "overestimated negative trends in national emission data bases". There is no indication about the version of the COPERT model used for the emission factor of transportation, although the last part of the investigated period is included in the "emission scandal". This point should be considered.

For the EURODELTA-Trends exercise, emissions officially reported by the countries were used. In the majority of European country, road transport emissions are estimated with the COPERT 4 model (Ntziachristos et al., 2009). This model includes detailed transport sources, fuel distribution, mileage, and level of penetration of control measures information. Emission factors used in the COPERT 4 model have been derived in the framework of several research projects conducted by various research institutes in Europe (Ntziachristos et al., 2009).
Recently, high uncertainty in $NO_x$ emissions, especially from the diesel vehicles, might be associated to non-compliance with air quality regulations. Several studies showed a substantial discrepancy (around a factor of 2-4) between different driving conditions (e.g. between the New European Driving Cycle and several other laboratory tests) in the $NO_x$ emission from light-duty diesel vehicles (Alves et al., 2013; Hausberger, 2010; May et al., 2013; Martin Weiss et al., 2011; M. Weiss et al., 2011). In addition, Karl et al., 2017 and Vaughan et al., 2016 also indicated a possible underestimation of real-word traffic emissions by investigating $NO_x$ flux measurements. Therefore, since NOx emission might have higher uncertainties than previously thought, the sentence at line 12 of page 11 was removed.

**Minor comments**

p. 5, l . 33: wrong parenthesis "can be found in Colette et al. (2017)"

Corrected.

p. 9, l. 38: why exactly 7 µg/m3 was chosen? Please explain

We removed measurements sites that exceeded 7 µg m$^{-3}$ (average over the full measurement periods) since we think that these values might be highly influenced by local pollution events which are not included in the underlying emission inventory. Out of the sites available in Tsimpidi et al., 2016, 3 stations were excluded. We included this information at line 39 of Page 9 of the manuscript. The threshold of 7 µg m$^{-3}$ is arbitrary, but choosing similar values of  5 µg m$^{-3}$  or  9 µg m$^{-3}$  would only slightly / not affect the results.

p.11 sub-paragraph 3.2.1: Please organize it in order along with the title "...SO2 and NO2 concentration trends": therefore it should be organized with relative trends for SO2 first and then relative trends for NO2. This is already done for the second part of the sub-paragraph about the significance.

Done.

p.12, l. 15: full stop is missing ": : : influenced by different factors. First : : :"

Corrected.

p.15 l. 21: ASOA and BSOA acronym were defined only in the abstract and not in the main text

We added the definition of the acronyms also in the main text at line 21 of page 15 of the revisited manuscript.

p. 16 l. 3: misspelling terpene as singular? "i.e. isoprene and terpenes"

Corrected.

p. 16 l. 18: misspelling of plays "could also play"

Corrected.

p. 17 l. 37: maybe you meant "latter" instead of "later"

Corrected.

p. 18 l. 13: maybe you meant "monoterpenes"?

Corrected.

p. 18 l. 14: this is the first comment about the influence of the change in Land Use, it is better to mention this in the discussion paragraph

The sentence has been re-phase at line 14 of page 18 of the revisited manuscript as below to make it clearer that we were referring to the underling land use data adopted by the different biogenic models and not to the change of land use itself in time.

*"The increase was independent of the land-use and specific biogenic model used and was mainly attributed to the increase of the surface temperature during the 1990–2010 period."*

Range of y-axis for figure 14 and 15 maybe set to 4 µg/m$^3$, actually figure 15 could be reduced to 3 µg/m$^3$: this would slightly improve its readability.

We prefer to keep the range y-axis as currently presented in the manuscript in order to (i) not over-lap the graphic with the legend in Figure 14 and to (ii) allow the readers for an immediate comparison between summer and winter absolute levels of the SOA concentrations presented in Figure 15.

Thanks for your suggestions. We revised the manuscript with all the points as addressed above individually.

**References**

Alves, C., Calvo, A., Lopez, D., Nunes, T., Charron, A., Goriaux, M., Tassel, P., Perret, P., 2013. Emissions of Euro 3-5 Passenger Cars Measured Over Different Driving Cycles, Int. J. Environ. Chem. Ecol. Geol. Geophys. Eng., 78, 294–297.

Boylan, J.W., Russell, A.G., 2006. PM and light extinction model performance metrics, goals, and criteria for three-dimensional air quality models. Atmospheric Environment 40, 4946–4959. https://doi.org/10.1016/j.atmosenv.2005.09.087.

Hausberger, S., 2010. Fuel consumption and emissions of modern passenger cars, TU Graz, 494, 1–8.

Karl, T., Graus, M., Striednig, M., Lamprecht, C., Hammerle, A., Wohlfahrt, G., Held, A., von der Heyden, L., Deventer, M.J., Krismer, A., Haun, C., Feichter, R., Lee, J., 2017. Urban eddy covariance measurements reveal significant missing NOx emissions in Central Europe. Sci Rep 7, 2536. https://doi.org/10.1038/s41598-017-02699-9.

May, A.A., Levin, E.J.T., Hennigan, C.J., Riipinen, I., Lee, T., Collett, J.L., Jimenez, J.L., Kreidenweis, S.M., Robinson, A.L., 2013. Gas-particle partitioning of primary organic aerosol emissions: 3. Biomass burning: BIOMASS-BURNING PARTITIONING. J. Geophys. Res. Atmos. 118, 11,327-11,338. https://doi.org/10.1002/jgrd.50828.

Ntziachristos, L., Gkatzoflias, D., Kouridis, C., Samaras, Z., 2009. COPERT: A European Road Transport Emission Inventory Model Information Technologies in Environmental Engineering 4th International ICSC Symposium, Thessaloniki, Greece, 2009, 491–504.

Tsimpidi, A.P., Karydis, V.A., Pandis, S.N., Lelieveld, J., 2016. Global combustion sources of organic aerosols: model comparison with 84 AMS factor-analysis data sets. Atmos. Chem. Phys. 16, 8939–8962. https://doi.org/10.5194/acp-16-8939-2016.

Vaughan, A.R., Lee, J.D., Misztal, P.K., Metzger, S., Shaw, M.D., Lewis, A.C., Purvis, R.M., Carslaw, D.C., Goldstein, A.H., Hewitt, C.N., Davison, B., Beevers, S.D., Karl, T.G., 2016. Spatially resolved flux measurements of NO x from London suggest significantly higher emissions than predicted by inventories. Faraday Discuss. 189, 455–472. https://doi.org/10.1039/C5FD00170F.

Weiss, M., Bonnel, P., Hummel, R., Manfredi, U., Colombo, R., Lanappe, G., Le Lijour, P., Sculati, M., 2011. Analyzing on-road emissions of light-duty vehicles with Portable Emission Measurement Systems (PEMS), JRC Scientific and Technical Reports, EUR, 24697.

Weiss, Martin, Bonnel, P., Hummel, R., Provenza, A., Manfredi, U., 2011. On-Road Emissions of Light-Duty Vehicles in Europe. Environ. Sci. Technol. 45, 8575–8581. https://doi.org/10.1021/es2008424.

**Responses to the comments of anonymous referee #2**

Thank you for your comments that helped to improve our manuscript. Please find below your comments in black, our responses in blue and modifications in the revised manuscript in *italic*.

The manuscript investigates trends in aerosol concentrations and chemical composition as well as their precursors in Europe over two decades that have experienced significant changes in emissions. The analysis is based on output of a suite of air quality models and observational data. The paper is well written and figures and tables are clear and informative. The paper contributes to the field of models evaluation and sensitivity analysis to different parameterizations and settings and will be of interest to GMD readers. However the following major points and specific comments could be addressed to improve it.

We thank the referee for the comments on the manuscript.

• The EURODELTA experiment includes output from eight air quality models of which five are included in the current manuscript. Although the additional three models do not allow a proper quantification of the trends they could be included as a model intercomparison exercise. Those models may indeed adopt more sophisticated (or computationally demanding) settings whose skills need to be quantified to better inform model users on the optimal model setup (see more details comments below). I would suggest including a new model intercomparison and evaluation section and present multiple statistical metrics of model skill for the years when all eight models have data.

We thank the referee for this comment. Since the final delivery of the EURODELTA-Trends model output data (Colette et al., 2017), two models had to withdraw from the final analysis due to a shortcoming in the model set-up, i.e. the WRF-Chem and the POLYPHEMUS model, therefore reducing the number of available models for the evaluation. However, we do agree with the referee and we included an evaluation for all the models and species available in the EURODELTA-Trends exercise. We have included a summary of the quality of the simulations for $NO_2$, $SO_2$, $SO_4^{2-}$, $TNO_3$ and $TNH_x$, as also suggested by the first reviewer. We calculated the mean fractional bias (MFB) and mean fractional errors (MFE) for all the models and species included in the exercise and for the two different periods (P1 and P2). The results were compared with the model performance criteria and performance goal proposed by Boylan and Russell, 2006.
Results indicated that the model performance criteria (MFB ≤ ±60 %, MFE ≤ +75 %) was met for all the species and by all the models that participated in the exercise, whereas the performance goal (MFB ≤ ±30 %, MFE ≤ +50 %) differs for some of the models and species. For instance, $TNO_3$ goal performance was not met by EMEP MSC-W in the P1 period and by CHIMERE for the P1 and P2 periods. Also, $SO_2$ performance goal was not met by MATCH during the P2 period and by MINNI in both the P1 and P2 period (Figure S4, below).

[Figure]

[Figure]

Figure S4: Soccer-goal plots for yearly concentrations of SO₂, SO₄²⁻, NO₂, TNO₃ and TNHₓ for the 1990–2000 (P1) period (left) and 2000–2010 (P2) period (right). Rows; from top to bottom: EMEP MSC-W, CHIMERE, MATCH, LOTOS-EUROS, MINNI. MFB: mean fractional bias; MFE: mean fractional error.

We added the following paragraph at line 1 of page 12 of the revisited manuscript and as well as Figure S4 in the revisited supplementary material.

*Model performance for SO₂ and NO₂ was additionally evaluated by calculating the mean fractional bias (MFB) and mean fractional error (MFE) for both the P1 and P2 periods separately (Appendix A). Recommended model performance criteria (MFB ≤ ±60 %, MFE ≤ +75 %) as well as the performance goal (MFB ≤ ±30 %, MFE ≤ +50 %) proposed by (Boylan and Russell, 2006) were achieved in both periods by most of the models apart for SO₂ for MINNI during the P1 and P2 period and for MATCH during the P2 period where only the model performance criteria were achieved (Figure S4).*

and the following paragraph at line 20 of page 13 of the revisited manuscript:

*Model performance for SO₄²⁻, TNO₃ and TNHₓ was also satisfactory; the recommended model performance criteria (MFB ≤ ±60 %, MFE ≤ +75 %) as well as the performance goal (MFB ≤ ±30 %, MFE ≤ +50 %) proposed by Boylan and Russell, 2006 were achieved in both the P1 and P2 periods by most of the models apart for TNO₃ in CHIMERE during the P1 and P2 period and in EMEP MSC-W during the P1 period where only the model performance criteria were achieved (Figure S4).*

•     Despite the paper focuses on the quantification of temporal trends, it is important to quantify the actual model skills vs observations. This could be useful also in the context of informing users on the optimal model setup to be adopted when aiming to simulate aerosol concentration and composition in future studies. It would be also useful to add results of these analyses to the conclusions, where the model/settings with highest skills are highlighted.

We agree with the referee and we included an evaluation for all the models and species investigated in the exercise, as presented in the previous comment. However, we preferred not to summarize them in the conclusion part since it is already quite lengthy, and we would like to emphasize the discussion on the ability of the models in reproducing the observed trends (and summarize them in the final part of the manuscript).

•     The trend analysis are performed for different aerosol species and over different regions. However there is no clear quantification of the spatial variability of these trends and how the model performance varies by year. It could be interesting to expand the current analyses and include a more detailed investigation on changes of model performance over different years and different sites. This will also contribute to a better quantification of inter-model variability and identification of sources of uncertainty in model output.

We agree with the referee and we performed an evaluation of the evolution of the single models mean fraction bias (MFB) over the full 1990–2010 period for all the investigated species. Results indicated that the mean fractional bias did not change substantially over the investigated period apart from $SO_2$ and $TNH_x$ where most of the models tend to be slightly more positively biased in 2010 compared to 1990 (Figure S5, below).

[Figure]

[Figure]

[Figure]

[Figure]

[Figure]

Figure S5: Evolution of the mean fractional bias (MFB) for $SO_2$, $SO_4^{2-}$, $NO_2$, $TNO_3$ and $TNH_x$ over the 1990–2010 period.

We added the following paragraph at line 6 of page 12 of the revisited manuscript and as well as Figure S5 on the revisited supplementary material.

*In addition, the evolution of the MFB over the full 1990–2010 period does not indicate any substantial change in 2010 compared to the first year of the exercise (i.e. 1990) with the exception of $SO_2$ which tends to be slightly more positively biased in the latter part of the period compared to 1990 (apart from EMEP MCS-W, Figure S5).*

and the following paragraph at line 24 of page 13 of the revisited manuscript:

*In addition, the evolution of the MFB over the full 1990–2010 period does not indicate any substantial change in 2010 compared to the first year of the exercise (i.e. 1990) apart from $TNH_x$ which tends to be slightly more positively biased in the latter part of the period compared to 1990 (Figure S5).*

On the other hand, the analysis of the spatial trends is complicated by the fact that most of the observational sites are mainly located in the Northern part of the domain, which limits the validation of the trend analysis to those sites. The latter has been noted also by the first reviewer. For example, for $TNO_3$ and $TNH_x$ almost the complete set of available stations is located in the Fennoscandia region, whereas no long-term measurements were available in the IP (Iberian Peninsula) as well as in FR (France)

for none of the species, therefore limiting the validation of the spatial variability of the trends. However, we have reported and discussed the model trends analysis in Figure 10 and Figure 13 of the manuscript for the PRUDENCE zones presented in Figure 1.

**Specific comment**

- Despite there is no word limit for GMD abstracts, the current one is very long, so I would suggest to present the results in a more concise way (e.g. by grouping models/species having similar performance/behavior, etc.).

We agree with the referee that the abstract is quite lengthy. However, due to the multiple topics investigated in the paper (e.g. trends in both secondary and inorganic aerosols as well as their gas-phase precursors), it is challenging to reduce the current length of the abstract. We therefore preferred to keep the abstract as in his current state.

- Page 2, line 7: I would use "evaluate" instead of "validate" here and in other instances in the paper since you are performing a model evaluation (not validation).

We agree and corrected also the other occurrences in the manuscript.

- Page 2, line 13: the meaning of relative trend or relative reduction (here) was never defined in the paper. It should be clarified here and/or in the methods since this terminology is used in many instances in the text ad figures.

We agree with the referee and we added the following paragraph at line 31 of page 9 of the revisited manuscript:

*The linear trends are presented as relative changes respect to the year 1990 and 2000 for the two 11-year periods, and as relative changes respect to the year 1990 for the full 21-year period.*

- Page 4, line 16: the years of the AeroCom experiment could be included

Done at line 17 of page 4 (the year is 2006).

- Page 5, line 24: what is the spatial resolution of the EDT models? Is it the same for all of them? This information should be included.

All the models had the same spatial resolution, i.e. 0.25° × 0.40°. We added this information at line 35 of page 5 of the revisited manuscript.

- Page 7, line 36: LAI from MODIS is available only from 2002. What did the EDT simulations use for prior years?

For the previous years, the last available year, i.e. 2002, was used for the LAI.

- Page 8, line 15: it would be interesting to have some more details about the temporal variability of emissions since they are one of the key drivers of changes in aerosol properties. This would also support your statement that trends in emissions are non-linear and would be helpful to interpret the results at finer temporal scales. Some information about the how trends in emissions change in space would be also relevant (you could also refer to literature studies if available).

We agree with the referee and we added the following information and reference about the spatial variability of the emission trends in $NO_2$ at line 15 of page 10 of the revisited manuscript:

*"with larger reductions occurring in Russia, Ukraine, Germany and the UK (Theobald et al., 2019)."*

and at line 11 of page 10 of the revisited manuscript for $SO_2$:

*"with larger reductions occurring in Germany and Eastern parts of the domain (Theobald et al., 2019)."*

Trend in emissions of $NO_2$ and $SO_2$ are reported in Table 2 of the manuscript. The analysis was divided in two periods 1990-2000 (P1) and 2000-2010 (P2) to show that larger declines in the gas-phase aerosol precursors occurred in the P1 period compared the P2 period.

- Page 9, line 27: this claim could be supported by a supplementary figure or referring to the literature, as suggested in a prior comment.

We agree with the referee and we added the following paragraph and reference at line 15 of page 10 of the revisited manuscript as described in the previous comment:

*"with larger reductions occurring in Russia, Ukraine, Germany and the UK (Theobald et al., 2019)."*

and at line 11 of page 10 of the revisited manuscript:

*"with larger reductions occurring in Germany and Eastern parts of the domain (Theobald et al., 2019)."*

- Page 10, line 5-16: these claims are not supported by your analyses. Could you show some more analyses and explain how the trends vary in space (or refer to the literature)?

Following our previous replies, we agree with the referee and we added the following paragraph and reference at line 15 of page 10 of the revisited manuscript:

*"with larger reductions occurring in Russia, Ukraine, Germany and the UK (Theobald et al., 2019)."*

and at line 11 of page 10 of the revisited manuscript:

*"with larger reductions occurring in Germany and Eastern parts of the domain (Theobald et al., 2019)."*

- Page 11, line 20: Period is missing at the end of the sentence. Also it is not clear the number of stations, data availability and frequency of the data used for the analysis.

The trend analysis was applied to the annual observational datasets. We added this information at line 3 of page 9 of the revisited manuscript as also suggested by the first referee as below:

*"The annual observational datasets […]"*

The number of stations for each species is reported in Section 2.3.4. of the revisited manuscript.

- Page 12, line 6: is there a way for you to indicate which dots these sites correspond to in Figure 7?

Different symbols could be used to refer to different stations. However, this will render the plots quite difficult to read. We prefer to keep the plot as it is now.

-      Page 13, line 6: do you have observations to be used for model evaluation in Figure 9/10?

For the purpose of this study, i.e. evaluating the trends in the nitrogen specie, we preferred to use measurements of total nitrate ($TNO_3$), mainly because the sampling of aerosol-nitrate can be problematic due to of evaporative loss of the semi-volatile ammonium nitrate or adsorption of nitric acid gas (Schaap et al., 2004). In addition, observation of nitric acid ($HNO_3$) are generally scare and quite challenging to perform.

-      Page 13, line 24: this claim could be better supported by referring to the spatial patterns of emissions, as suggested to do in previous comments.

We agree with the referee previous suggestions and we added the following paragraph and reference at line 15 of page 10 of the revisited manuscript as described in the previous comment:

*"with larger reductions occurring in Russia, Ukraine, Germany and the UK (Theobald et al., 2019)."*

And at line 11 of page 10 of the revisited manuscript:

*"with larger reductions occurring in Germany and Eastern parts of the domain (Theobald et al., 2019)."*

**Tables**

Table 7: there is no mention of Table 7 in the manuscript but I would keep it, so please add reference to it and comments. Also the statistical metrics should be defined. What is the temporal resolution/frequency of the data used in this evaluation and how many observations do you have at each site?

We correct the cross-reference at line 15 of page 15 with Table 7. The mean values have been taken from the work of Tsimpidi et al., 2016 and are based on aerosol mass spectrometer (AMS) measurements performed at hourly resolution. However, the number of observations available at each site is not directly reported in the study (i.e. the average values provided in the study were used for the evaluation). We agree with the referee and we added the definition of the statistical metrics in the Appendix A of the revisited manuscript.

**Figures**

Figure 5 and 8: not clear how many observational sites are included and the temporal resolution of the data used.

We added the number of observational sites in the caption of the updated version of Figure 5 and Figure 8. The trend analysis was applied to the annual observational datasets.

Figure 14: Rephrase caption to clarify that panel a includes all seasons as in Table S2, while summer and winter campaigns are shown in panels b and c.

We re-phrased the caption of Figure 14 as below:

*"Modeled and observed (retrieved from PMF analysis) means and standard deviations of SOA concentrations for all periods (a), for summer campaigns (b) and winter campaigns (c) (Table S2)."*

Figure S1: why averages above 7 μg m-3 are excluded? Explain this also in Table S1.

We removed measurements sites that exceeded 7 ug m$^{-3}$ (average over the full measurement periods) since we think that these values might be highly influenced by local pollution events which are not included in the underlying emission inventory. We indicated this information is Table S2 of the supplementary material. Out of the sites available in Tsimpidi et al., 2016, 3 stations were excluded. We included this information at line 39 of Page 9 of the manuscript. The threshold of 7 μg m$^{-3}$ is arbitrary, but choosing similar values of 5 μg m$^{-3}$ or 9 μg m$^{-3}$ would only slightly / not affect the results.

Thanks for your suggestions. We revised the manuscript with all the points as addressed above individually.

**Technical corrections:**

- Page 2, line 34: be consistent between "emission" and "emissions".

Corrected.

- Page 3, line 10: remove "the formation of".

Removed.

- Page 4, line 29: SIA was not defined earlier.

We added the definition (secondary inorganic aerosol).

- Page 8, line 35: typo "Weather Research and Forecasting".

Corrected.

- Page 9, line 11: typo "illustrates".

Corrected.

- Page 12, line 24: replace "model" with "models".

Done.

- Page 17, line 5: add also spatial resolution (if it is the same).

Done.

- Page 17, line 21: SOx missing subscript.

Corrected.

- Figure 2: Missing subscripts on titles for chemical species.

Corrected.

**Reference**

Boylan, J.W., Russell, A.G., 2006. PM and light extinction model performance metrics, goals, and criteria for three-dimensional air quality models. Atmospheric Environment 40, 4946–4959. https://doi.org/10.1016/j.atmosenv.2005.09.087.

Colette, A., Andersson, C., Manders, A., Mar, K., Mircea, M., Pay, M.P., Raffort, V., Tsyro, S., Cuvelier, C., Adani, M., Bessagnet, B., Bergström, R., Briganti, G., Butler, T., Cappelletti, A., Couvidat, F., D'Isidoro, M., Doumbia, T., Fagerli, H., Granier, C., Heyes, C., Klimont, Z., Ojha, N., Otero, N., Schaap, M., Sindelarova, K., Stegehuis, A.I., Roustan, Y., Vautard, R., van Meijgaard, E., Vivanco, M.G., Wind, P., 2017. EURODELTA-Trends, a multi-model experiment of air quality hindcast in Europe over 1990–2010. Geoscientific Model Development 10, 3255–3276. https://doi.org/10.5194/gmd-10-3255-2017.

Schaap, M., Spindler, G., Schulz, M., Acker, K., Maenhaut, W., Berner, A., Wieprecht, W., Streit, N., Müller, K., Brüggemann, E., Chi, X., Putaud, J.-P., Hitzenberger, R., Puxbaum, H., Baltensperger, U., ten Brink, H., 2004. Artefacts in the sampling of nitrate studied in the "INTERCOMP" campaigns of EUROTRAC-AEROSOL. Atmospheric Environment 38, 6487–6496. https://doi.org/10.1016/j.atmosenv.2004.08.026.

Theobald, M.R., Vivanco, M.G., Aas, W., Andersson, C., Ciarelli, G., Couvidat, F., Cuvelier, K., Manders, A., Mircea, M., Pay, M.-T., Tsyro, S., Adani, M., Bergström, R., Bessagnet, B., Briganti, G., Cappelletti, A., D'Isidoro, M., Fagerli, H., Mar, K., Otero, N., Raffort, V., Roustan, Y., Schaap, M., Wind, P., Colette, A., 2019. An evaluation of European nitrogen and sulfur wet deposition and their trends estimated by six chemistry transport models for the period 1990–2010. Atmospheric Chemistry and Physics 19, 379–405. https://doi.org/10.5194/acp-19-379-2019.

Tsimpidi, A.P., Karydis, V.A., Pandis, S.N., Lelieveld, J., 2016. Global combustion sources of organic aerosols: model comparison with 84 AMS factor-analysis data sets. Atmos. Chem. Phys. 16, 8939–8962. https://doi.org/10.5194/acp-16-8939-2016.

**Responses to the short comment of Prof. Dr. David Ham**

Thank you for your short comments that helped to improve our manuscript. Please find below your comments in black, our responses in blue and modifications in the revised manuscript in *italic*.

The code and data availability section in this manuscript fails to comply with GMD requirements for transparency and archiving in a number of ways. These need to be remedied before the manuscript can be accepted.

**1 EMEP data**

The documentation cited here is on a wiki server. This is an entirely ephemeral location which could change at any time. It is important that the information required to replicate the data are persistently available. The website pointed to is quite short, so one way to achieve this would be to reformat the content as a short report which could be archived in one of the following ways:

1.       **As an appendix to the paper.**
2.       **On a preprint server such as arXiv.**
3.       **On a persistent data repository such as https://zenodo.org**

Care needs to be taken that this information is comprehensive (i.e. everything needed to rerun the experiment has been specified) and that the references to data stored on the THREDDS server are sufficiently precise that the exact data can be retrieved. Ideally this would be by attaching persistent identifiers such as DOIs to the data and referring to these.

The THREDDS server also seems to be inaccessible. Are these really the correct citations?

We fully agree we your comments and we uploaded all the EMEP observational data as well as the R procedures on a persistent data repository (https://doi.org/10.5281/zenodo.3405386). The link has been added in the Data and Code availability section of revisited version of the manuscript. Additionally, data available on the THREDDS server have been made available on the AeroCom server under the */metno/aerocom-users-database/EURODELTA/* folder, and such information provided in the Data and Code availability section.

**2 Results data**

The results data section just links to a wiki page which is mostly about how to use ssh. No mechanism is provided for identifying the results data. Please ensure that the location of the results data is sufficiently precisely identifed that the reader could trace back from the results presented in the paper to the data from which those results were computed. Once again, please ensure that this information is in a persistent archived location (in particular not a wiki page or other website) so that there can be some confidence that future readers will be able to access this information.

We agree with your comment and we provided additional information of the specific location of the model input and output and how to obtain the data in the Data and code availability section of the revisited manuscript as shown below.

*"Technical details of the EURODELTA project simulations that permit the replication of the experiment are available on the wiki of the EMEP Task Force on Measurement and Modelling (https://wiki.met.no/emep/emep-experts/tfmmtrendeurodelta, last access: 21 December 2018), which also includes ESGF links to corresponding input forcing data. The EURODELTA-Trends model results are made available for public use on the AeroCom server (information to gain access to the AeroCom server are available at https://wiki.met.no/aerocom/ user-server, last access: 21 December 2018). Models input and output data are permanently stored under the /metno/aerocom-users-database/EURODELTA folder on the AeroCom Server. See Colette et al. (2017) for full terms and conditions for the use of these data. Measurements data and the R procedures are available online at: https://doi.org/10.5281/zenodo.3405386."*

**3 Evaluation code**

Providing code on request does not satisfy GMD's requirements for transparency and reproducibility. The reasons for this are given in detail in the new editorial https://doi. org/10.5194/gmd-12-2215-2019. Please therefore provide a persistent, public archive of the evaluation code, for example on https://zenodo.org.

We agree we your comments and we provided the evaluation code on a persistent data repository (https://doi.org/10.5281/zenodo.3405386). The link has been added in the Data and Code availability section of revisited version of the manuscript as presented in the previous comment.

Thanks for your suggestions. We revised the manuscript with all the points as addressed above individually.